# How accurate are operational dust models in predicting Particulate Matter (PM) levels in the Eastern Mediterranean Region? Insights from PM Surface Concentrations

Andreas Eleftheriou[1], Petros Mouzourides[1], Panayiotis Kouis[2], Nikos Kalivitis[3], Itzhak Katra[4], Emily Vasiliadou[5], Chrysanthos Savvides[5], Panayiotis Yiallouros[2], Marina K.-A. Neophytou[1]

[1]Environmental Fluid Mechanics Laboratory, Department of Civil and Environmental Engineering, University of Cyprus, Nicosia, Cyprus

[2]Medical School, Cyprus University, Cyprus

[3]Department of Chemistry, University of Crete, Greece

[4]Department of Environmental, Geoinformatics, and Urban Planning Sciences, Ben-Gurion University of the Negev, Israel

[5]Department of Labour Inspection, Ministry of Labour and Social Insurance, Cyprus

*Correspondence to*: Petros Mouzourides (pmouzou@ucy.ac.cy)

**Abstract.** This study provides the first comprehensive assessment of eleven operational dust forecast models and a Multi-model Median (MMM) in predicting ground-level Particulate Matter (PM) concentrations in the Eastern Mediterranean Region (EMR), with a focus on Cyprus, Greece, and Israel. Ground-based observations from regional background stations support model performance assessment across different PM fractions ($PM_{10}$, $PM_{2.5}$, and coarse particles), using established statistical metrics (correlation coefficient, R, Mean Bias, MB, and Root Mean Square Error, RMSE). Across the evaluation subsets, full period, high-dust conditions (95th percentile), and confirmed dust-day events, model skill varies considerably by site and particles fraction (R ranging from −0.24 to 0.91). NASA-GEOS exhibits the highest overall correlations across sites (R = 0.71, 0.65, and 0.64 at Ayia Marina, Be'er Sheva, and Finokalia, respectively), while NOA-WRF demonstrates better performance during intense dust events, achieving R = 0.91 at Finokalia (95th percentile). The MMM reduces biases and errors relative to many individual models, providing stable performance across subsets. Performance generally improves for the coarse particle fraction, and BOOT and categorical diagnostics indicate reduced scatter and clearer hit rates during dust-dominated days. However, no single model performs optimally across all sites and conditions; spatial heterogeneity (proximity to sources/orography) and configuration choices (vertical resolution, first-layer height, horizontal resolution, and data assimilation) drive these differences. The study underscores the importance of refining model configurations and improving parameterizations to enhance forecast accuracy. Future efforts should incorporate localized data and further develop region-specific models to improve the operational use of these systems in early warning protocols for mitigating public health impacts.

## 1. Introduction

Desert Dust Storm (DDS) events have been documented since ancient times, with early references found in texts such as "Histories" by Herodotus (Herodotus III: 86-88). These events impact human health and infrastructure, yet they remain understudied until recent decades.

Over the past few years, research into DDSs has expanded rapidly. Recognising their global impacts, the United Nations General Assembly designates 2025-2034 as the Decade for combatting DDSs (United Nations, 2024). This growing attention reflects the mounting evidence linking DDSs to respiratory and cardiovascular illnesses (Lorentzou et al., 2019) and mental health effects

(Jones, 2023), regardless of the source region (Lwin et al., 2023). In parallel, climate change, population growth, and geopolitical instability increasingly exacerbate desertification, driving more frequent and intense DDSs and reinforcing transboundary dust transport patterns (Eleftheriou et al., 2023). Therefore, monitoring both the source and downwind impacts of DDSs remains critical for assessing and mitigating their effects.

To address air quality concerns, European directives (e.g. 2004/107/EC, 2008/50/EC, and 2015/1480/EU) and the World Health Organization (WHO) air quality guidelines (2021), establish threshold values to regulate pollution from both anthropogenic and natural sources. These regulatory frameworks enable authorities to track elevated PM concentrations from human activities, while also capturing high PM levels during natural events such as DDSs. During such episodes, PM concentrations often exceed 150–200 µg/m³, well above regulatory limits. This capability supports timely public warnings and facilitates health-protective measures when air quality deteriorates due to dust intrusions. The integration of monitoring networks with operational forecast models, such as METAL-WRF (Solomos et al., 2023), enhances our understanding of DDS dynamics and supports compliance with air quality regulations. These systems also strengthen community resilience, providing predictive capacity to mitigate both anthropogenic and natural air pollution. In this context, Early Warning Systems (EWSs) play a key role in protecting the public during hazardous natural events, including DDSs.

EWSs play a critical role in protecting populations from climate-related and natural hazards, including DDSs. EWSs help reduce mortality and economic losses associated with extreme weather and hydrometeorological events. However, major implementation gaps persist, particularly in vulnerable regions such as small island developing states and least-developed countries. According to the United Nations, only 50% of countries operate adequate multi-hazard EWSs, despite the fact that 70% of disaster-related deaths over the past 50 years occurred in the most vulnerable areas. Bridging this gap by advancing forecasting technologies and ensuring broader access to EWSs remains essential, especially for managing transboundary phenomena like DDSs, where timely and accurate forecasts can significantly reduce health and environmental impacts.

The accuracy of EWSs ultimately depends on the performance of dust forecasting models, which vary considerably in their configuration and resolution. These models differ in first-layer calculation heights, ranging from approximately 20 to 100 meters above ground or sea level and in the particle size ranges they consider, spanning 0.03 to 20µm (Knippertz and Stuut, 2014). They also incorporate diverse emission schemes (e.g., Marticorena and Bergametti, 1995) and deposition processes (e.g., Zender et al., 2003). These differences directly influence model outputs and contribute to substantial variability in dust concentration forecasts. Model evaluations commonly rely on sun photometers (e.g., AERONET; Holben et al., 1998) or satellite-derived aerosol optical depth (AOD) products (e.g., MODIS; NASA, 2024), which primarily capture column-integrated dust loads rather than near-surface concentrations. Over time, satellite detection algorithms have significantly improved, incorporating geostationary sensors in addition to traditional polar-orbiting platforms (e.g., Kolios and Hatzianastassiou, 2019). Unlike polar-orbiting satellites, geostationary platforms provide continuous temporal coverage and are not constrained by the same revisit-time limitations, thereby enhancing the monitoring of dust transport and diurnal variability. These approaches, while valuable, are subject to inherent limitations. For example, cloud cover can significantly reduce the accuracy of satellite retrievals, while the low temporal resolution of polar-orbiting satellites constrains their ability to capture rapid dust transport dynamics and event-scale variability (Kazadzis et al., 2009). Nevertheless, ongoing initiatives, such as the Horizon Europe CiROCCO project, are designed to address these limitations by developing integrated predictive frameworks and improving both the spatial and temporal coverage of dust monitoring and forecasting systems.

Evaluating dust surface concentrations forecasts using ground $PM_{10}$ concentrations is also challenging, as it is difficult to isolate the desert dust fraction in these measurements (Garcia-Castrillo & Tarradellas, 2017). Consequently, very few studies have evaluated numerical models using data from near-ground monitoring stations, and, to the best of our knowledge, none have focused

on the Eastern Mediterranean Region (EMR). This region lies at the crossroads between Africa and Middle East, both of which are major sources of transboundary dust pollution. The study of the EMR has become increasingly difficult in the past decade due to ongoing conflicts and socio-political instability in the nearby dust source regions. This instability has limited in-situ data collection and delayed any mitigation efforts for DDSs (Eleftheriou et al., 2023).

In this study, we evaluate surface dust concentration forecasts from eleven (11) operational dust models and their multi-model median (MMM) using daily ground-level PM measurements from three background stations in the EMR. These stations, Ayia Marina (AM; Cyprus), Finokalia (FKL; Greece), and Be'er Sheva (BS; Israel), offer low-background environments, allowing for reliable assessment of long-range transboundary dust transport. Observed PM data are categorized into $PM_{10}$, $PM_{2.5}$, and coarse particle fractions, and we apply a suite of statistical metrics to assess model performance. The measurements include 24-hour averages of both observed PM and modelled dust surface concentrations, ensuring temporal alignment. The performance evaluation incorporates different subsets of the data, including the entire study period, the 95th percentile of PM concentrations, and dust storm days, as classified using the methodology of Achilleos et al. (2020). Section 2 describes the EMR context and the selected study sites. Section 3 outlines data sources, model configurations, and the statistical evaluation methods. Section 4 presents results across all evaluation scenarios and PM fractions, including graphical diagnostics such as Taylor diagrams, BOOT methodology, and contingency tables. Section 5 summarizes findings and discusses implications for model refinement and operational forecasting.

The aim of this work is to determine how accurately current operational models forecast surface dust levels in the EMR, using background PM observations as reference. To our knowledge, this represents the first large-scale, multi-model evaluation of its kind for the region, which is critically exposed to dust intrusions but remains underrepresented in model validation studies. The findings highlight key model strengths and limitations and inform future efforts to improve dust prediction in support of health-focused EWS.

## 2. Characterizing DDSs in the EMR: Sources and Monitoring

The Mediterranean Basin frequently experiences DDSs, particularly in its southern and eastern regions, which are strongly influenced by emissions from the Sahara and the Sahel (Querol et al., 2009). Recent studies employing combined satellite and ground-based observations have provided a more detailed picture of these events. For instance, Gkikas et al. (2016), using an algorithm that integrates multi-sensor satellite data with in-situ measurements, found that strong dust storms occur more frequently throughout the year in the western Mediterranean Basin, whereas the most intense events tend to develop in the central region. The study also identified a clear seasonal pattern: dust storms are more common during summer in the western Mediterranean, while they predominantly occur in spring across the central and eastern parts. Earlier work by Gkikas et al. (2013) similarly demonstrated that the spatiotemporal characteristics of Mediterranean dust storms are more effectively captured through satellite-based retrievals, emphasizing the key role of remote sensing in characterizing regional dust dynamics. In addition, Salvador et al. (2022) reported a statistically significant upward trend in the occurrence of African Dust Outbreaks and their intensity over the period 1948–2020. These findings indicate an increase in the frequency of air mass transport from North Africa toward the western Mediterranean Basin and a corresponding intensification of dust episodes during recent decades.

Using air parcel back-trajectory analysis, Varga et al., (2014) further showed that dust frequently enters the Eastern Mediterranean Region (EMR) from North Africa, highlighting the dominant role of African sources in regional dust intrusions. Numerous studies report significant air quality degradation during these events in areas such as mainland Greece, the North Aegean, Cyprus, Israel, and Turkey (Çapraz et al., 2021; Triantafyllou et al., 2020; Tsiflikiotou et al., 2019; Vratolis et al., 2019; Krasnov et al., 2016; Mouzourides et al., 2015). In addition to African sources, dust emissions from the Middle East significantly affect the EMR. For example, Bodenheimer et al. (2019) analysed 53 DDS events, with concentrations exceeding 150 μg/m³ using data from 13 air

quality monitoring stations in Israel (2007–2013). Their findings confirm the transboundary and multi-source nature of dust episodes, which often combine contributions from North African and Middle Eastern source regions.

Figure 1 illustrates the study area and background monitoring sites. These include Ayia Marina Xyliatou (AM) in Cyprus and Finokalia (FKL) in Crete, both part of the European Monitoring and Evaluation Programme (EMEP), and Be'er Sheva (BS) in Israel, part of the Israeli National Air Monitoring Network. These stations are strategically located in relatively isolated areas, minimizing local anthropogenic influence and making them ideal for evaluating long-range dust transport across the EMR.

Each station continuously records PM concentrations using high temporal resolution instruments: the Tapered Element Oscillating Microbalance (TEOM) for Cyprus and Israel, and the FH 62 I-R Thermo analyser for Greece. Measurements are collected at heights of 2–5 m above ground level at AM and FKL, and 10–15 m above ground at BS, where instruments are installed on building rooftops. These setups ensure consistent and representative sampling of ambient air, particularly for coarse PM fractions, which are most affected by dust intrusions. Furthermore, monitoring stations record particle sizes of $PM_{10}$ and $PM_{2.5}$.

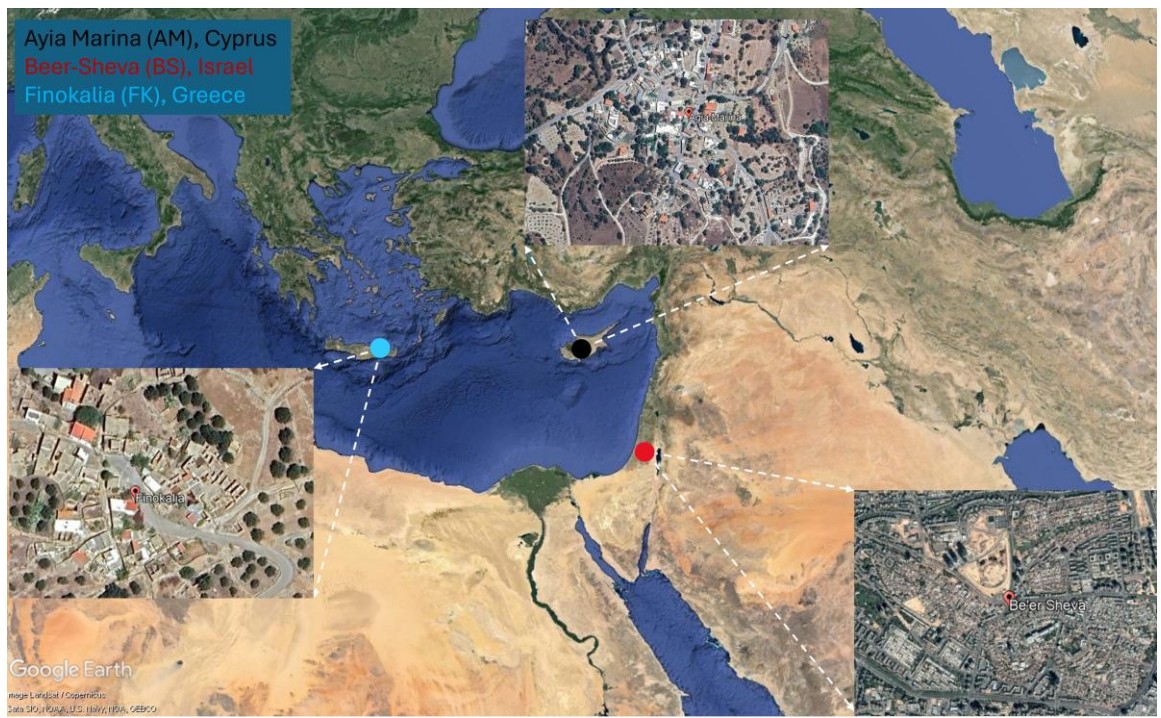

Figure 1: Study area and monitoring sites in the Eastern Mediterranean. Locations of Ayia Marina Xyliatou (AM; Cyprus) , Finokalia (FKL; Greece) and Be'er Sheva (BS; Israel) used in the model-observation evaluation. The stations are strategically positioned away from major anthropogenic emission sources, making them suitable for evaluating transboundary desert dust transport. Imagery is obtained from © Google Earth 2025.

In this study, we analyse daily averaged forecasts from 11 operational dust models available through the SDS-WAS platform, now known as the WMO Barcelona Dust Regional Center (https://sds-was.aemet.es/). Additionally, we use a MMM ensemble, which represents the median surface dust concentration predicted by all participating models for each site. Table 1 and Table 2 summarise the main configuration characteristics of each model, including meteorological drivers, emission schemes, and spatial resolution, providing a consistent basis for performance intercomparison.

We compare modelled surface dust concentrations to daily observations of $PM_{10}$ and $PM_{2.5}$ collected at the three sites. Before comparing with model outputs, the ground-based PM measurements underwent additional processing to ensure data consistency and reliability. This involves applying quality control checks to remove outliers and erroneous values, converting high-frequency data into 24-hour means, and harmonizing timestamps across sites to correct for time zone and logging differences. These steps are essential for minimizing bias and enabling a robust model–observation comparison.

The observation period covers 2012–2018 for AM, 2012–2017 for BS, and 2012–2016 for FKL. $PM_{2.5}$ data are not available for FKL during this timeframe. Data collection and processing follow the MEDEA project protocol (Achilleos et al., 2020; 2023), co-funded by the LIFE 2016 Programme, which aims to evaluate indoor exposure reduction interventions during DDSs. As part of this effort, Achilleos et al. (2020) develop a dust classification methodology that combines satellite imagery, $PM_{10}/PM_{2.5}$ ratios, elemental composition (Ca, Al, Fe), and dust aerosol optical depth (Dust-AOD) to standardize dust storm identification across all three sites. Using this multi-criteria approach, the authors identify 106 dust storm days at AM, 88 at FKL, and 101 at BS.

Figure 2 illustrates the daily $PM_{10}$ evolution at the three regional-background stations (AM with black, FKL with blue, BS with red) which serves as the basis for the subsequent performance analyses. The overall concentration levels are relatively low throughout the study period, generally below 50 µg/m³, confirming the regional background character of these sites. Intermittent high-concentration episodes do occur at all locations. Be'er Sheva exhibits the highest variability, with several sharp peaks exceeding 600 µg/m³. At AM, moderate peaks reach up to ~400–500 µg/m³. Finokalia, in contrast, displayed the lowest mean concentrations and the most stable temporal trend, with minimal outlier behaviour. Overall, the three stations represent consistent regional background conditions for the Eastern Mediterranean Basin yet exhibit clear spatial contrasts in the intensity and frequency of episodic dust intrusions.

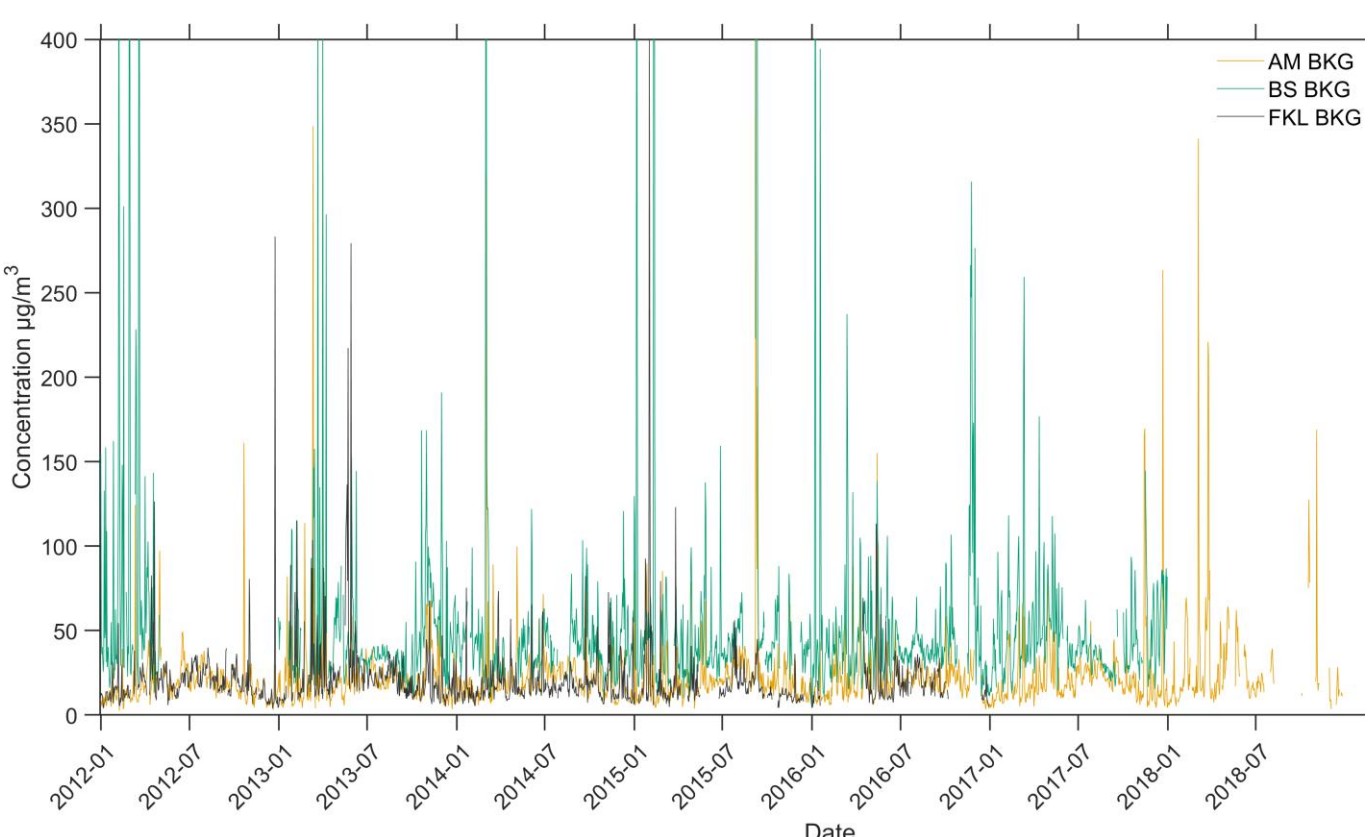

Figure 2: Daily mean $PM_{10}$ concentrations (µg/m³) at the three monitoring stations used for model evaluation, 2012–2018: Ayia Marina, Cyprus (AM, orange), Be'er Sheva, Israel (BS, teal), and Finokalia, Crete (FKL, grey). The series show predominantly low background levels (<50 µg m⁻³) punctuated by episodic high-dust events; more frequent and intense at BS and intermittent at AM, while FKL remains comparatively stable.

Table 1: Configuration details of the 11 dust forecast models included in this study. These configurations correspond to the versions used during the study period (2012–2018) and may differ from current operational settings.

| Model | Institution | Meteorological driver | Regional or global coverage | Meteorological initial fields | Radiation interactions | Emission scheme |
|---|---|---|---|---|---|---|
| **BSC_DREAM 8b_V2** | BSC | Eta/NCEP | Regional | NCEP/GFS | Dust–radiation feedbacks | Uplifting (Shao et al., 1993; Janjic et al., 1994; Nickovic et al., 2001) |
| **MACC-ECMWF** | CAMS | ECMWF | Global | ECMWF/IFS | No | Uplifting (Ginoux et al., 2001; Morcrette et al., 2008, 2009) |
| **DREAM8-MACC** | SEEVCCC | NMME | Regional | ECMWF/IFS | No | Uplifting (Shao et al., 1993; Janjic et al., 1994) |
| **NMMB-BSC** | BSC | NMMB/NCEP | Regional | NCEP/GFS | Interactive dust-radiation coupling | Saltation and sandblasting (Janjic et al., 1994; Marticorena and Bergametti, 1995) |
| **NASA_GEOS** | NASA | GEOS-5 | Global | GEOS-5 Analysis | Direct effects fully included | Based on Ginoux (2001) |
| **NCEP_NGAC** | NCEP | NEMS GFS | Global | NCEP/GDAS | Yes (not active) | Dust uplifting following Ginoux (2001) |
| **EMA_REG4** | EMA | RegCM4 | Regional | NCEP/GFS | Yes (both short and long waves) | Saltation and sandblasting (Zakey et al., 2006; Marticorena and Bergametti, 1995; Alfaro and Gomes, 2001) |
| **DREAMABOL** | CNR-ISAC | BOLAM | Regional | NCEP/GFS | Not active | Uplifting (Tegen et al., 1994) |
| **NOA_WRF** | NOA | WRF | Regional | GFS | No | GOCART scheme by Ginoux et al. (2001) |
| **SILAM** | FMI | ECMWF | Global | Offline Model | Photolysis rates are dependent on overlaying PM in our global 0.2 x 0.2 forecast with chemist | Sofiev et al., 2015 |
| **LOTOS** | TNO | ECMWF | Regional | ECMWF | No | Marticorena & Bergametti 1995. Dust uplifting following Shao et al. (2001). |

Table 2: Output configuration of the dust models used in the performance evaluation. These characteristics influence dust dispersion representation and forecast accuracy at ground level.

| Model | Horizontal resolution | Vertical resolution | Height first layer | Transport size bins | Data assimilation | Reference |
|---|---|---|---|---|---|---|
| **BSC_DREAM 8b_V2** | 1/3º x 1/3º | 24 Eta-layers | 86 m (above sea level) | 8 bins (0.1-10μm) | No | Nickovic et al., 2001; Pérez et al., |

| | | | | | | 2006; Basart et al., 2012 |
|---|---|---|---|---|---|---|
| **MACC-ECMWF** | 8-10 km approx. (O1280) | 137 sigma-layers | 10 m (above surface) | 3 bins (0.03-20μm) | Yes AOD550/MODIS | Rémy et al., 2022 |
| **DREAM8-MACC** | 1/3º x 1/3º | 24 Eta-layers | 96 m (above sea level) | 8 bins (0.1-10μm) | Yes (ECMWF dust-analysis) | Nickovic et al., 2016 |
| **NMMB-BSC** | 1/3º x 1/3º | 24 sigma-hybrid-layers | 108 m (above surface) | 8 bins (0.1-10μm) | No | Pérez et al., 2011; Klose et al., 2021 |
| **NASA_GEOS** | 0.25º x 0.3125º | 72 layers (Top: 0.01 hPa) | Dynamic from 20 to 100 meters | 5 bins (0.73-8.0 μm) | Yes (MODIS) | Colarco et al., 2010 |
| **NCEP_NGAC** | T126 (~ 1º) | 64 sigma-pressure hybrid layers (Top at 0.2 hPa) | 20 m | 5 bins (0.73-8.0 μm) | No | Lu et al., 2016 Zhang et al., 2022 |
| **EMA_REG4** | 45 km x 45 km | 18 sigma-layers | 50 m | 4 bins (0.01-20) | No | Zakey et al., 2006 |
| **DREAMABOL** | 0.4º x 0.4º | 50 sigma-hybrid levels | 27 m above surface | 8 bins (0.1-10μm) | No | Micrea et al., 2008; Mauruizi et al., 2011 |
| **NOA_WRF** | 0.19º x 0.22º | 40 vertical levels | 20 | 5 bins (0.73-8μm) | No | Flaounas et al., 2017 |
| **SILAM** | 0.1 x 0.1 deg (dust only) 0.2 x 0.2 deg (version with all modeled aerosols and full model chemistry) | 19 hybrid levels up to about 16 km (0.1 x 0.1 dust only simulation) 28 hybrid levels up to about 50 km (0.2 x 0.2 simulation) | 10m | 4 bins (0.1-30 μm) | No | Sofiev et al., 2015 |
| **LOTOS** | 0.50∘longitude × 0.25∘ latitude | 12-15 layers | 25 m above surface level | 5 bins (0.1-10μm) | No | Manders et al., 2017 |

## 3. Evaluation Methodology and Performance Metrics

To evaluate the performance of the studied models, we use different statistical metrics, including the Pearson correlation coefficient (R), Mean Bias (MB), and Root Mean Square Error (RMSE). These metrics capture different aspects of model accuracy: R quantifies the strength of the linear relationship between observed and predicted values (ranging from −1 to 1), MB indicates systematic over- or underestimation, and RMSE reflects the overall magnitude of error, with greater sensitivity to large discrepancies. The evaluation is carried out by comparing model outputs with observational data from the three background stations located in Cyprus, Greece, and Israel. This comprehensive assessment of dust models in the Eastern Mediterranean Region (EMR), to the best of our knowledge has never been done before and provides valuable insights into their performance in an understudied area.

### 3.1. Statistical Performance Metrics for Model Evaluation

The R coefficient measures the linear relationship between modelled and observed values, ranging from -1 to 1. A value of 1 indicates a perfect positive correlation, -1 indicates a perfect negative correlation, and a value of 0 implies no correlation. This metric is crucial for identifying whether the patterns of observation data and forecasts align.

$$R = \frac{1}{(n-1)} \sum_{i=1}^{n} \left( \frac{M_i - \bar{M}}{\sigma_M} \right) \left( \frac{O_i - \bar{O}}{\sigma_O} \right) \tag{1}$$

where $M_i$ and $O_i$ are the predicted and observed values respectively, and $n$ is the number of observations.

Mean Bias ($MB$) is used to determine the average deviation of predicted values from observed values. A positive $MB$ indicates that the model overestimates the observed values, while a negative MB indicates underestimation. This metric is significant as accurate forecasting can help vulnerable populations take precautionary measures during DDS events. The MB is defined as:

$$MB = \frac{1}{n} \sum_{i=1}^{n} M_i - O_i \tag{2}$$

The Root Mean Square Error (RMSE) measures the differences between predicted and observed values and is useful for indicating the model's overall prediction error. RMSE penalizes larger errors more heavily than smaller ones, making it a valuable metric for assessing model performance. It is defined as:

$$RMSE = \left( \sum_{i=1}^{n} \frac{(M_i - O_i)^2}{n} \right)^{1/2} \tag{3}$$

While these metrics effectively summarize model performance, they also have limitations. Specifically, the $R$ and $RMSE$ are linear measures and may fail to capture nonlinear relationships between observed and predicted values. Moreover, although $RMSE$ penalizes larger errors, it remains sensitive to extreme outliers, which potentially skewing the evaluation results, especially during intense dust events. Recognizing these limitations is essential for interpreting the results and identifying areas where models may need further refinement. To enhance the comparative analysis among models, a ranking system was introduced in the performance tables to indicate which models perform best across different sites and statistical metrics.

### 3.2. Graphical Representations: Taylor Diagrams, BOOT Evaluation and Contingency Tables

Additionally, we employed the Taylor diagram, which provides a graphical summary of the three (3) metrics of $R$, $RMSE$ and the Standard Deviation ($SD$) in a single plot. Beyond summarizing performance metrics, the Taylor diagram offers a clear and intuitive means of comparing different models' performances against observational data. In this visualization the $SD$ is proportional to the radial distance from the origin, while $RMSE$ corresponds to the distance from the "observed" point on the x-axis. The green dashed contours indicate specific values and the $R$ is represented along the arc of the diagram, increasing from the top toward the bottom, as illustrated in Figure 3, for example.

The BOOT Model Evaluation Methodology (Chang & Hanna, 2004) is also used, incorporating bootstrapping techniques to resample the data and assess model performance in terms of underestimation or overestimation compared to observed values. The BOOT methodology was chosen over other resampling techniques, such as cross-validation, due to its flexibility and robustness in handling smaller datasets and its ability to provide a statistically sound assessment of model scatter and bias. Bootstrapping allows for the recreation of relationships between observed and modelled data using smaller, resampled datasets (Lahiri, 2010), providing a statistically robust comparison of model performance, and has been used in several recent studies (Lumet et al., 2024; Yang et al., 2024). This methodology uses two key metrics: Geometric Mean ($MG$) a measure of relative bias and the Geometric Variance ($VG$) a measure of relative scatter, with the perfect model positioned at the bottom centre of the graph.

$$MG = exp(\overline{lnO_t} - \overline{lnM_t}) \tag{4}$$

$$VG = exp\left( \overline{lnO_t - lnM_t}^2 \right) \tag{5}$$

We evaluated model skill using categorical statistics that focus on how well models capture observations rather than just their magnitude. Specifically, we built a contingency table that counts, for each day, how often the model overestimates, underestimates, or accurately predicts PM concentrations—where "accurate" means the model falls within ±1 SD of the ground observations. The daily heatmaps use our traffic-light binning for PM levels (0–50, 50–100, 100–200, and >200 µg/m³), and the cell values show the percentage of comparisons in each condition (over, under, within-1SD) relative to the total number of model–observation pairs for that day.

### 3.3. Approaches for Assessing Model Prediction against Observations

In this study, we evaluate model performance using multiple data subsets and comparative approaches to capture different aspects of forecast skill. The observed data include the PM$_{10}$ ($O_{PM_{10}}$), PM$_{2.5}$ ($O_{fine}$) and $PM_{coarse}$ ($O_{coarse}$) observations. The latter is calculated as the difference between PM$_{10}$ and PM$_{2.5}$. For the model predictions, only dust concentration data ($M_{dust_{conc}}$) is used. We apply two primary comparison strategies:

   i.    Modelled dust concentration against PM$_{10}$ observations ($M_{dust_{conc}}$ vs $O_{PM_{10}}$),
   ii.   Modelled dust concentration against $PM_{coarse}$ observations ($M_{dust_{conc}}$ vs $O_{coarse}$).

These approaches enable evaluation of total particulate loading and the coarse-mode fraction, which is more directly linked to mineral dust. Each comparison is performed across three temporal evaluation scenarios:

   i.    the entire study period for each region,
   ii.   the 95$^{th}$ percentile of observed daily PM concentrations, representing extreme events, and
   iii.  identified dust storm days, based on the Achilleos et al. (2020), classification framework, which incorporates criteria such as PM$_{10}$ exceedances, Dust Aerosol Optical Depth (Dust-AOD), MODIS satellite imagery, PM ratios, and elemental crustal signatures (Ca, Al, Fe).

This methodology follows the structure proposed by García-Castrillo & Terrandellas (2017) but extends it by incorporating the BOOT evaluation technique (see Section 3.2). To our knowledge, this is the first multi-model validation effort in the Eastern Mediterranean Region (EMR) to apply this combination of methods across multiple PM fractions, locations, and dust-intense periods. For clarity and completeness, we exclude outliers in BOOT visualizations that fall outside the evaluation graph boundaries, as these indicate very low predictive skill and would disproportionately affect interpretability.

### 4. Assessment of model performance

In this section, we present a comprehensive evaluation of model performance based on the statistical and graphical methods described in Section 3. The results are organized around the three evaluation scenarios: (a) the entire study period, (b) the 95$^{th}$ percentile of observed PM concentrations, and (c) the subset of dust storm days identified using the Achilleos et al. (2020) criteria. Each model's ability to reproduce observed ground-level PM concentrations is assessed using PM$_{10}$ and $O_{coarse}$ data across all sites. Results are reported using statistical metrics ($R$, $MB$, and $RMSE$) and visual diagnostics, including Taylor diagrams, the BOOT methodology, and categorical heatmaps based on contingency analysis. This multi-metric framework allows for a nuanced comparison of model skill, emphasizing both trend representation and bias magnitude under different dust-loading conditions.

#### 4.1. Evaluation of Model Performance Under Different Scenarios

#### 4.1.1. Results for the Entire Study Period

Table 3 and Figures 3–4 present the comparative evaluation of eleven operational dust forecasting models and the MMM ensemble against observed surface $O_{PM_{10}}$ and $O_{coarse}$ concentrations. Overall, model performance varies substantially among sites, reflecting

the combined influence of emission parameterizations, meteorological drivers, and local environmental conditions across the EMR. Most models reproduce the temporal variability of dust concentrations with moderate correlations ($R \approx 0.4 - 0.7$) but show systematic biases and site-dependent errors. NASA-GEOS and the MMM ensemble perform best overall, achieving the highest correlations ($R \geq 0.56$) and relatively balanced bias profiles. In contrast, EMA-REG4 and LOTOS-EUROS consistently underperform, with weak correlations ($R \lesssim 0.3$) in most sites and high RMSE values, reaching up to 172 µg/m³ and 144 µg/m³ respectively, at Be'er Sheva.

Distinct spatial patterns emerge across the monitoring network. At AM and FKL, most models underestimate PM$_{10}$ concentrations, with MB values typically between −5 and −15 µg/m³, whereas BS shows consistent overestimation, with MB values frequently between 10 and 40 µg/m³. The highest RMSE values are recorded at BS (70–110 µg/m³), reflecting the site's proximity to active desert dust sources and its strong day-to-day variability. In contrast, lower RMSE values are found at FKL (20–40 µg/m³) and AM (25–35 µg/m³), where observed concentrations represent well-mixed regional background conditions. The best-performing systems, i.e NASA-GEOS, NOA-WRF, and MMM, combine correlation coefficients exceeding 0.6 with relatively low absolute biases ($|MB| \leq 15\,\mu g/m^3$), indicating robust and stable performance across all metrics. Intermediate results are obtained for the DREAM model family (DREAMABOL, DREAM8-MACC, and DREAM8b_V2), with correlations between 0.40 and 0.63, RMSE values between 25 and 85 µg m⁻³, and alternating bias signs depending on site and particle size fraction.

The use of $O_{coarse}$ instead of $O_{PM_{10}}$ generally reduces model bias, particularly at AM, where MB improves from −9.84 µg/m³ to 0.66 µg/m³ for the MMM ensemble and from −13.87 µg/m³ to −3.43 µg/m³ for DREAM8-MACC. However, RMSE values remain within similar ranges (20–80 µg/m³), indicating that coarse-fraction filtering mitigates mean offset but does not eliminate amplitude errors in event representation. BS continues to show the largest spread in MB and RMSE values, with positive MB up to 37 µg/m³ and RMSE exceeding 100 µg/m³, emphasizing the persistent difficulty of simulating near-source dust uplift.

Analysis of the BOOT methodology (Figure 4) supports these conclusions, showing that well-performing models such as NASA-GEOS, NOA-WRF, and MMM cluster closer to the zero-bias line and exhibit reduced scatter, particularly for Be'er Sheva and Ayia Marina. Conversely, low-performing systems like EMA-REG4, LOTOS-EUROS, and SILAM show points outside the confidence boundaries, indicating substantial deviations in both magnitude and variability. Despite minor inconsistencies between BOOT plots and traditional performance metrics for specific models, the overall trends are consistent, confirming the robustness of the comparative evaluation framework.

In summary, across the full evaluation period, NASA-GEOS and the MMM ensemble are the most consistent performers (R > 0.6, RMSE $\lesssim$ 75 µg m⁻³ overall), with NOA-WRF particularly strong at Be'er Sheva (R $\approx$ 0.7, RMSE = 106 µg m⁻³). EMA-REG4 shows the weakest predictive skill (R ≤ 0.3, RMSE > 170 µg m⁻³), while LOTOS-EUROS exhibits mixed behaviour—performing moderately well at regional background sites but losing skill near dust-source regions. NCEP shows the opposite tendency, correlating better near sources but with positive bias. Better agreement at FKL (mean $R \approx 0.55$, $RMSE \approx 25\,\mu g/m^3$) indicates that models reproduce long-range transport more accurately than near-source processes. Overall, the ensemble median (MMM) reduces individual model variability and provides the most balanced representation of $O_{PM_{10}}$ and $O_{coarse}$ concentrations, underscoring its utility for operational dust forecasting in the EMR.

Table 3: Ranked statistical performance (correlation coefficient *R*, mean bias *MB*, and root mean square error *RMSE*) of the eleven operational dust-forecast models and the MMM against observed PM$_{10}$ ($O_{PM_{10}}$) and coarse fraction (O$_{coarse}$) at Ayia Marina (AM), Be'er Sheva (BS), and Finokalia (FKL) over the full study period. Table 3a summarizes results for $O_{PM_{10}}$, and Table 3b for O$_{coarse}$. Rankings are by performance of R coefficient

| Model | AM (R) | AM$_{RMSE}$ (µg/m³) | AM$_{MB}$ (µg/m³) | | BS (R) | BS$_{RMSE}$ (µg/m³) | BS$_{MB}$ (µg/m³) | | FKL (R) | FKL$_{RMSE}$ (µg/m³) | FKL$_{MB}$ (µg/m³) |
|---|---|---|---|---|---|---|---|---|---|---|---|
| NASA_GEOS | 0.71 | 35.07 | 3.6 | NASA_GEOS | 0.65 | 89.48 | 14.98 | NASA_GEOS | 0.64 | 41.24 | 0.64 |
| NOA_WRF | 0.56 | 38.38 | 7.41 | NOA_WRF | 0.62 | 105.99 | 36.54 | MMM | 0.64 | 20.43 | -10.88 |
| MMM | 0.56 | 23.55 | -9.84 | MMM | 0.58 | 71.79 | 13.74 | DREAMABOL | 0.63 | 19.72 | -7.25 |
| DREAM8_MACC | 0.48 | 26.05 | -13.87 | NMMB | 0.56 | 86.19 | -25.04 | NCEP | 0.62 | 29.24 | 3.4 |
| LOTOS-EUROS | 0.47 | 46.39 | -2.74 | MACC_ECMWF | 0.54 | 72.78 | -28.5 | DREAM8b_V2 | 0.58 | 24.08 | -10.32 |
| MACC_ECMWF | 0.47 | 26.56 | -11.84 | NCEP | 0.52 | 68.76 | 7.96 | NOA_WRF | 0.58 | 33.01 | 1.67 |
| NMMB | 0.43 | 28.17 | -15.03 | DREAM8_MACC | 0.51 | 79.26 | 0.57 | DREAM8_MACC | 0.54 | 21.93 | -13.55 |
| DREAMABOL | 0.42 | 27.53 | -4.04 | DREAM8b_V2 | 0.49 | 85.2 | 0.57 | MACC_ECMWF | 0.5 | 28.08 | -11.51 |
| DREAM8b_V2 | 0.4 | 27.16 | -7.93 | DREAMABOL | 0.49 | 69.35 | 3.66 | SILAM | 0.49 | 31.94 | -9.22 |
| SILAM | 0.38 | 61.42 | -7.31 | SILAM | 0.4 | 95.29 | -8.33 | NMMB | 0.42 | 25.54 | -14.91 |
| NCEP | 0.27 | 54.03 | 7.88 | EMA_REG4 | 0.25 | 172.4 | 16.35 | EMA_REG4 | 0.31 | 21.54 | -12.48 |
| EMA_REG4 | 0.02 | 34.27 | -19.54 | LOTOS-EUROS | 0.24 | 144.44 | -2.63 | LOTOS-EUROS | 0.23 | 13.11 | -12.33 |

(b)

| Model | AM (R) | AM$_{RMSE}$ (µg/m³) | AM$_{MB}$ (µg/m³) | | BS (R) | BS$_{RMSE}$ (µg/m³) | BS$_{MB}$ (µg/m³) | | FKL (R) | FKL$_{RMSE}$ (µg/m³) | FKL$_{MB}$ (µg/m³) |
|---|---|---|---|---|---|---|---|---|---|---|---|
| NASA_GEOS | 0.7 | 39.44 | 14.23 | NOA_WRF | 0.67 | 113.99 | 54.64 | – | – | – | – |
| NOA_WRF | 0.57 | 42.75 | 17.54 | NASA_GEOS | 0.6 | 100.09 | 36.11 | – | – | – | – |
| MMM | 0.56 | 18.46 | 0.66 | MMM | 0.56 | 76.47 | 34.78 | – | – | – | – |
| DREAM8_MACC | 0.47 | 17.96 | -3.43 | MACC_ECMWF | 0.51 | 60.59 | -7.83 | – | – | – | – |
| MACC_ECMWF | 0.46 | 20.83 | -1.39 | NMMB | 0.51 | 85.56 | -3.68 | – | – | – | – |
| LOTOS-EUROS | 0.45 | 47.3 | 8.21 | DREAM8_MACC | 0.49 | 80 | 21.84 | – | – | – | – |
| DREAMABOL | 0.42 | 25.95 | 6.32 | NCEP | 0.49 | 69.94 | 28.91 | – | – | – | – |
| NMMB | 0.42 | 20.78 | -4.46 | DREAMABOL | 0.48 | 65.61 | 23.31 | – | – | – | – |
| DREAM8b_V2 | 0.41 | 22.94 | 2.6 | DREAM8b_V2 | 0.46 | 92.51 | 56.95 | – | – | – | – |
| SILAM | 0.37 | 61.77 | 3.14 | SILAM | 0.41 | 94.88 | 8.73 | – | – | – | – |
| NCEP | 0.27 | 56.16 | 18.34 | LOTOS-EUROS | 0.3 | 143.73 | 13.91 | – | – | – | – |
| EMA_REG4 | 0.01 | 24.79 | -8.77 | NOA_WRF | 0.67 | 113.99 | 54.64 | – | – | – | – |

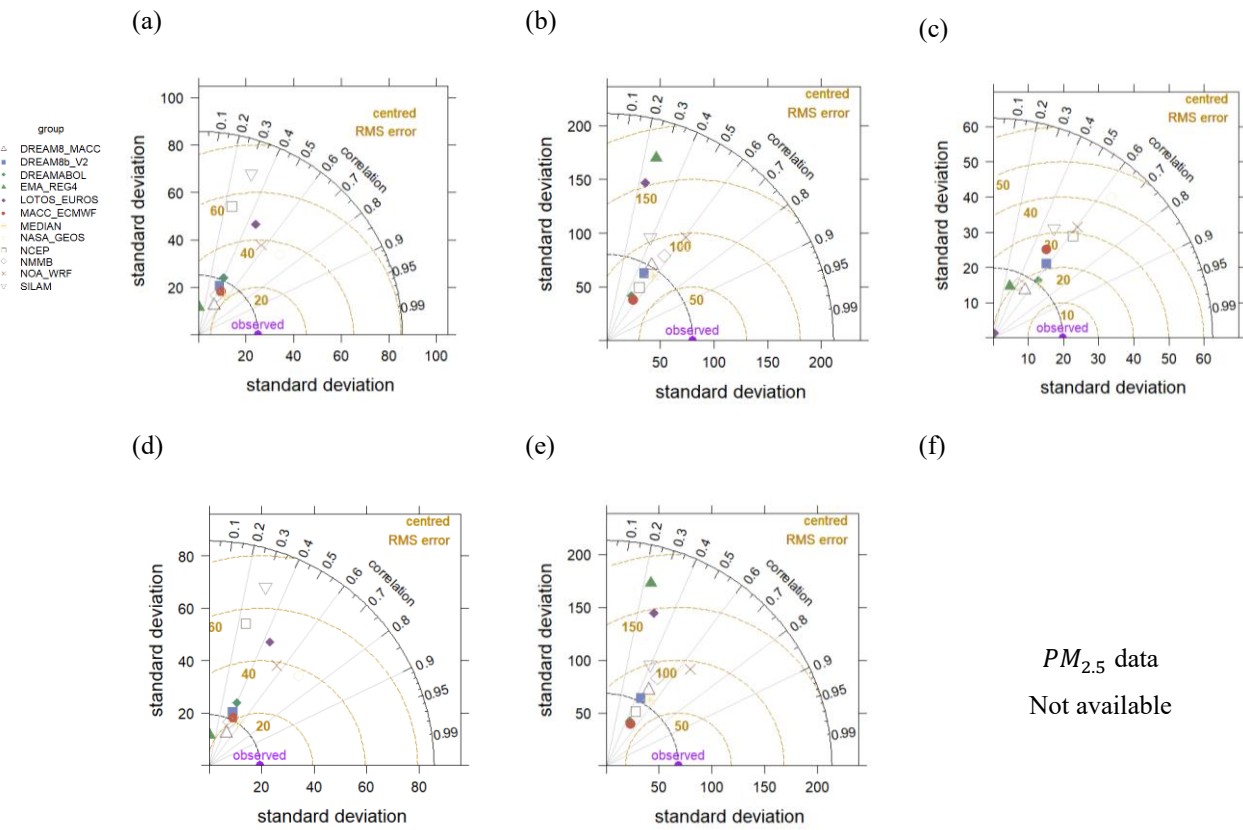

$PM_{2.5}$ data

Not available

Figure 3: Taylor diagrams for daily PM₁₀ over the full study period at the three monitoring sites. Panels show model performance against observations at (a and d) Ayia Marina (AM), (b and e) Be'er Sheva (BS), and (c) Finokalia (FKL). Panels (a–c) correspond to method (i) $M_{dust_{conc}}$ vs $O_{PM_{10}}$ and (d–e) correspond to method (ii) $M_{dust_{conc}}$ vs $O_{Coarse}$. The azimuth indicates the correlation

coefficient (R), the radial distance the standard deviation (SD), and the concentric arcs the centered RMSE. Symbols denote the 11 operational models; the multi-model median (MMM) is highlighted. The observational reference is located at R=1 and SD=1.

(a)                                                        (b)

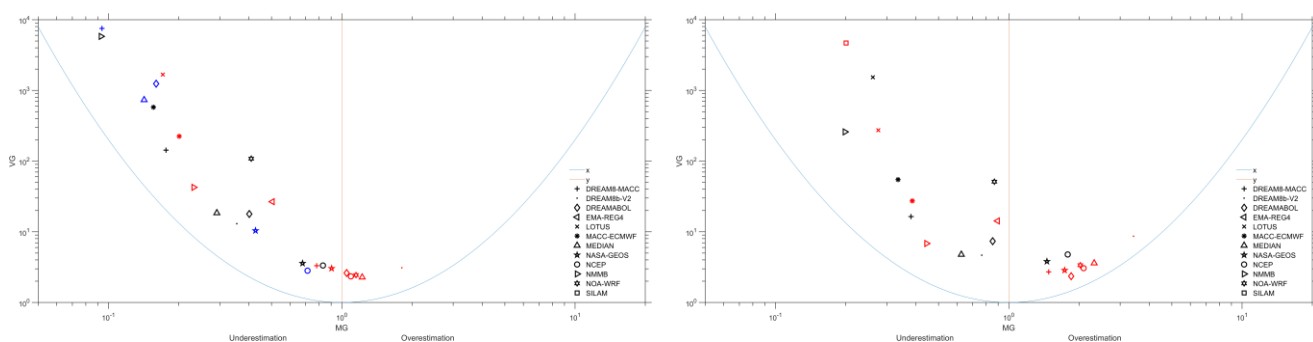

Figure 4: BOOT statistical plots comparing modelled daily PM₁₀ concentrations to observed values over the full study period at the three monitoring stations: Ayia Marina (AM), Be'er Sheva (BS), and Finokalia (FKL). a) $M_{dust_{conc}}$ vs $O_{PM_{10}}$ and b) $M_{dust_{conc}}$ vs $O_{Coarse}$. Models closer to the origin demonstrate better agreement with observations. Outliers beyond graph limits are excluded for visual clarity.

### 4.1.2.    Results for the 95th Percentile of Measurements

This analysis evaluates model performance in reproducing the most intense dust events, represented by the 95th percentile of daily mean PM concentrations. Overall, most models exhibit reduced skill compared to the full-period evaluation, highlighting the challenge of simulating extreme events where both emissions and transport processes become highly nonlinear. Nevertheless, clear contrasts among models and sites emerge. NASA-GEOS and NOA-WRF consistently rank among the best-performing systems, achieving the highest correlations and accurately capturing the timing of peak dust episodes across the EMR. NOA-WRF reaches

R = 0.91 at FKL, the highest correlation recorded and maintains good agreement at AM (R = 0.53) and BS (R = 0.55). Similarly, NASA-GEOS performs strongly at AM (R = 0.66) and FKL (R = 0.56), with slightly lower skill at BS (R = 0.49). These results indicate that both models effectively reproduce the temporal evolution of major dust outbreaks, particularly in coastal and inland environments strongly influenced by African and Middle Eastern sources. However, their positive mean biases, especially at BS (up to 150–200 µg/m³), demonstrate a systematic overestimation of event intensity. This behaviour suggests that while these models

capture the occurrence of dust storms well, they overpredict near-source emissions and coarse-particle loadings, resulting in elevated RMSE values exceeding 300 µg/m³.

A second group of models, including DREAMABOL, DREAM8-MACC, DREAM8b_V2, NCEP, and the MMM ensemble, show moderate skill, with correlations typically between 0.4 and 0.6 and mixed bias behaviour. For example, DREAMABOL and DREAM8b_V2 reproduce high-dust variability at FKL and BS but underestimate at AM, while NCEP and the MMM show more

balanced bias patterns, with moderate underestimation in Cyprus and mild overestimation in Israel. The MMM continues to provide robust, smoothed estimates of dust concentrations, reducing extremes and offering a stable compromise among individual model tendencies. Despite these advantages, RMSE values remain high (often above 250 µg/m³), revealing that even the ensemble approach struggles to match the magnitude of extreme dust concentrations.

In contrast, models such as EMA-REG4, NMMB, and SILAM show limited ability to reproduce severe dust events, with

correlations typically below 0.3–0.4 and RMSE exceeding 250–450 µg m⁻³. LOTOS-EUROS performs similarly overall but shows localized improvements, achieving R = 0.59 and RMSE = 54 µg m⁻³ at Ayia Marina, indicating occasional skill in long-range transport under specific high-dust conditions. The comparison between $O_{PM_{10}}$ and $O_{Coarse}$ simulations, provides additional insight. For many models, bias decreases when fine particles are excluded, indicating that misrepresentation of the fine fraction contributes significantly to the total PM₁₀ error. For instance, DREAM8-MACC at Ayia Marina improves from MB = −54.97 to −28.81 µg/m³,

and NCEP at BS from −86.47 to −22.56 µg/m³ when using $O_{Coarse}$. However, improvements in RMSE are limited, showing that large errors in event magnitude persist even after accounting for size partitioning. These findings confirm that reproducing the intensity of extreme dust episodes remains a critical challenge, as current emission and transport schemes struggle to accurately represent the rapid onset and decay of strong outbreaks.

Table 4: Ranked performance statistics for the 95th percentile (high-dust) subset against observed PM10 ($O_{PM_{10}}$) and coarse fraction ($O_{coarse}$) at Ayia Marina (AM), Be'er Sheva (BS), and Finokalia (FKL). Table 3a summarizes results for $O_{PM_{10}}$, and Table 3b for $O_{coarse}$. Rankings are by performance of R coefficient for each site.

| Model | AM (R) | AMRMSE (µg/m³) | AMMB (µg/m³) | | BS (R) | BSRMSE (µg/m³) | BSMB (µg/m³) | | FKL (R) | FKLRMSE (µg/m³) | FKLMB (µg/m³) |
|---|---|---|---|---|---|---|---|---|---|---|---|
| NASA_GEOS | 0.66 | 113.29 | 41.01 | NOA_WRF | 0.55 | 282.49 | 163.83 | NOA_WRF | 0.91 | 66.98 | 37.45 |
| NOA_WRF | 0.53 | 78.54 | 25.96 | NASA_GEOS | 0.49 | 300.83 | 83.73 | DREAMABOL | 0.6 | 54.98 | −15.90 |
| LOTOS-EUROS | 0.34 | 108.52 | −20.79 | DREAM8b_V2 | 0.45 | 253.79 | −71.10 | NASA_GEOS | 0.56 | 127.04 | 44.96 |
| NCEP | 0.32 | 78.27 | −10.50 | NMMB | 0.44 | 315.37 | −67.19 | MMM | 0.55 | 56.38 | −25.42 |
| MMM | 0.32 | 80.27 | −35.41 | MMM | 0.42 | 252.92 | −46.16 | DREAM8_MACC | 0.53 | 64.07 | −42.74 |
| MACC_ECMWF | 0.23 | 90.59 | −43.39 | DREAM8_MACC | 0.4 | 263.76 | −61.93 | NCEP | 0.52 | 67.91 | 21.92 |
| DREAM8b_V2 | 0.22 | 89.8 | −42.32 | NCEP | 0.4 | 251.04 | −86.47 | DREAM8b_V2 | 0.51 | 63.21 | −14.46 |
| DREAM8_MACC | 0.2 | 93.3 | −54.97 | DREAMABOL | 0.39 | 274.44 | −112.28 | EMA_REG4 | 0.48 | 55.23 | −46.97 |
| NMMB | 0.18 | 96.58 | −53.53 | SILAM | 0.39 | 228.5 | 11.73 | MACC_ECMWF | 0.4 | 88.5 | −25.17 |
| DREAMABOL | 0.17 | 79.28 | −24.00 | MACC_ECMWF | 0.34 | 286.51 | −157.67 | NMMB | 0.27 | 81.93 | −50.52 |
| SILAM | 0.1 | 244.42 | 25.01 | LOTOS-EUROS | 0.33 | 240.86 | 27.73 | SILAM | 0.25 | 76.28 | 1.48 |
| EMA_REG4 | −0.24 | 117.84 | −90.34 | EMA_REG4 | 0.3 | 447.58 | −53.59 | LOTOS-EUROS | 0.23 | 13.11 | -12.33 |
| (b) | | | | | | | | | | | |
| Model | AM (R) | AMRMSE (µg/m³) | AMMB (µg/m³) | | BS (R) | BSRMSE (µg/m³) | BSMB (µg/m³) | | FKL (R) | FKLRMSE (µg/m³) | FKLMB (µg/m³) |
| NASA_GEOS | 0.63 | 128.02 | 68.06 | NOA_WRF | 0.6 | 302.59 | 197.87 | | – | – | – |
| LOTOS-EUROS | 0.59 | 54.19 | −3.94 | NASA_GEOS | 0.4 | 345.94 | 153.1 | | – | – | – |
| NOA_WRF | 0.51 | 87.22 | 48.4 | DREAM8b_V2 | 0.36 | 239.84 | −9.57 | | – | – | – |
| NCEP | 0.31 | 69.67 | 17.63 | NMMB | 0.35 | 324.01 | −7.00 | | – | – | – |
| MMM | 0.31 | 60.33 | −8.59 | MMM | 0.35 | 244.69 | 17 | | – | – | – |
| DREAM8_MACC | 0.2 | 68.7 | −28.81 | DREAM8_MACC | 0.34 | 253.08 | 2.52 | | – | – | – |
| MACC_ECMWF | 0.2 | 70.76 | −16.51 | DREAMABOL | 0.33 | 226.17 | −50.58 | | – | – | – |
| DREAM8b_V2 | 0.18 | 70.91 | −12.69 | SILAM | 0.33 | 234.93 | 43.18 | | – | – | – |
| NMMB | 0.15 | 75.86 | −25.29 | NCEP | 0.3 | 226.39 | −22.56 | | – | – | – |
| DREAMABOL | 0.1 | 69.74 | 4.25 | MACC_ECMWF | 0.28 | 244.58 | −97.75 | | – | – | – |
| SILAM | 0.05 | 245.77 | 56.54 | EMA_REG4 | 0.24 | 444.59 | 0.1 | | – | – | – |
| EMA_REG4 | −0.23 | 89.6 | −63.91 | LOTOS-EUROS | 0.2 | 325.55 | 98.75 | | – | – | – |

(a)  (b)  (c)

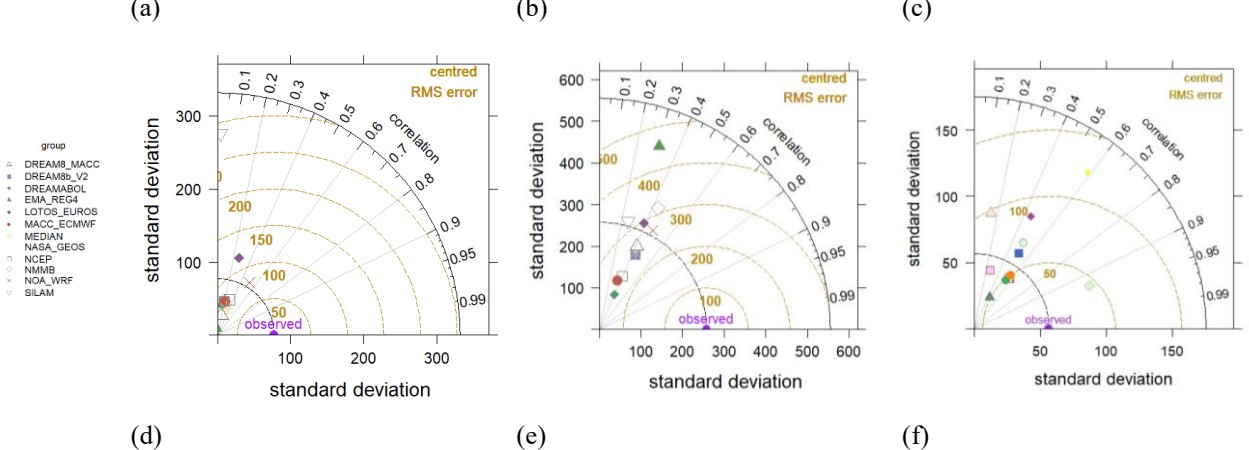

(d)  (e)  (f)

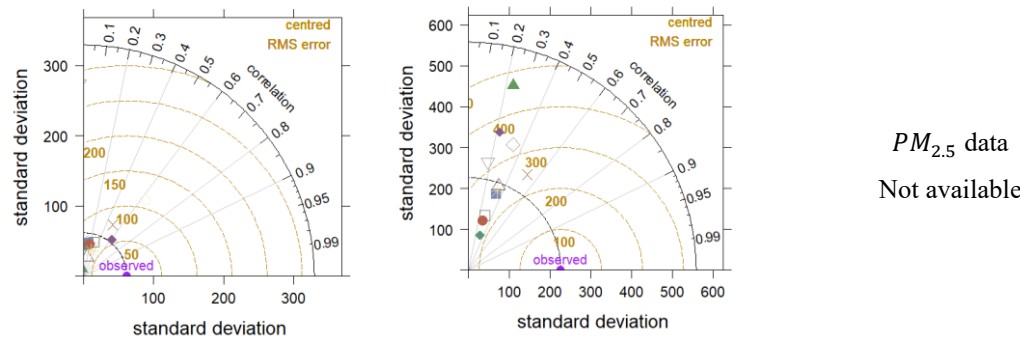

PM$_{2.5}$ data

Not available

Figure 5: Taylor diagrams showing the performance of 11 operational dust models and the multi-model median (MEDIAN) during the 95$^{th}$ percentile of observed PM$_{10}$ concentrations at the three monitoring stations. Panels show model performance against observations at (a and d) Ayia Marina (AM), (b and e) Be'er Sheva (BS), and (c) Finokalia (FKL). Panels (a–c) correspond to method (i) $M_{dust_{conc}}$ vs $O_{PM_{10}}$ and (d–e) correspond to method (ii) $M_{dust_{conc}}$ vs $O_{Coarse}$. This analysis reflects the ability of each model to reproduce the timing and magnitude of extreme dust events.

(a)                                                           (b)

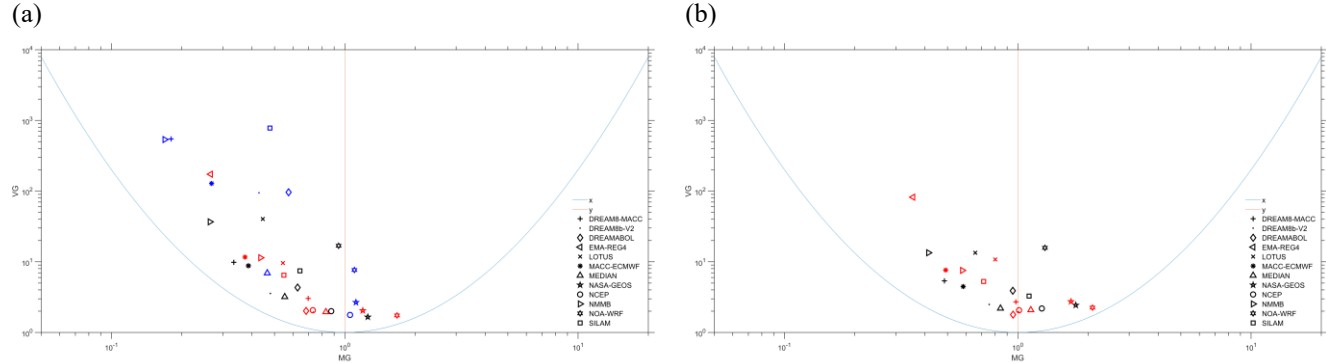

Figure 6: BOOT statistical plots comparing modelled and observed PM$_{10}$ concentrations for the 95$^{th}$ percentile of daily values at Ayia Marina (AM), Be'er Sheva (BS), and Finokalia (FKL). a) $M_{dust_{conc}}$ vs $O_{PM_{10}}$ and b) $M_{dust_{conc}}$ vs $O_{Coarse}$. Models positioned closer to the origin show better performance. Extreme outliers are excluded for readability.

### 4.1.3.    Results for Dust Days Identified by Achilleos et al. (2020) methodology

Figures 6 and 8 and Tables 5a–b summarize model skill on confirmed dust days identified by the Achilleos et al. (2019) methodology. Relative to the full-period results and the high-dust, rankings converge and inter-model spread is reduced, reflecting the stronger event focus of the dust-day selection.

NASA-GEOS and the MMM remain the most consistent performers, sustaining R ≥ 0.60 at all three sites with |MB| ≤ 40 µg/m³ in most cases and RMSE generally ≤ 75 µg/m³. NOA-WRF retains high event skill, especially at BS and FKL, with R frequently > 0.60 and competitive RMSE, consistent with its improved timing during intense episodes. Systems previously identified as lower-performing (EMA-REG4, LOTOS-EUROS) continue to exhibit weak correlations (R < 0.30) and large errors on dust days, although extreme outliers are fewer than in the 95$^{th}$ percentile analysis.

The dust-day approach and the 95$^{th}$ percentile filter yield broadly consistent rankings for the best (NASA-GEOS, NOA-WRF, MMM) and least-performing (EMA-REG4, LOTOS-EUROS) models. Notably, MACC-ECMWF and NCEP improve under the dust-day selection, moving up the ranks at one or more sites, indicating that multiparameter event identification (transport, optical/aerosol signals) better aligns model timing and vertical structure with observed peaks. Across models, the dust-day selection reduces bias scatter relative to the 95$^{th}$ percentile subset; for example, at AM, MMM MB improves from −9.84 to 0.66 µg/m³, and

DREAM8-MACC from -13.87 to -3.43 µg/m³ when using $O_{Coarse}$ rather than $O_{PM_{10}}$ (see §4.1.1–4.1.2), while RMSE remains in the 20–80 µg/m³ range, implying that mean offsets are mitigated more than amplitude errors.

BS continues to exhibit the largest spread in MB and RMSE (positive MB up to 37 µg/m³, RMSE > 100 µg/m³ for several systems), highlighting persistent challenges in simulating near-source uplift even when analysis is restricted to confirmed dust events. In contrast, FKL maintains tighter errors (typical RMSE ≈ 20–40 µg/m³) and R ≈ 0.5–0.6, consistent with long-range transport dominance. AM sits between these regimes, with moderate biases and correlations that closely track the all-days ranking.

BOOT plots for dust days (Figure 8) corroborate the metric-based ranking: NASA-GEOS, NOA-WRF, and MMM cluster near the zero-bias axis with reduced dispersion at AM and BS, whereas EMA-REG4, LOTOS-EUROS, and SILAM appear outside confidence bounds more frequently, indicating persistent magnitude and variability mismatches. Minor layout differences relative to the 95[th] percentile BOOT do not alter the overall conclusions.

On confirmed dust days, NASA-GEOS and the MMM deliver the most consistent skill (R typically ≥ 0.60, RMSE ≤ 45-110 µg m⁻³), NOA-WRF remains event-strong at BS/FKL, and MACC-ECMWF/NCEP show notable gains versus the 95[th] percentile analysis. The dust-day selection reduces ranking dispersion and bias variability, reinforcing the robustness of the comparative evaluation framework.

Table 5: Ranked performance statistics for dust days as defined by Achilleos et al. (2019) against observed PM₁₀ ($O_{PM_{10}}$) and coarse fraction ($O_{coarse}$) at Ayia Marina (AM), Be'er Sheva (BS), and Finokalia (FKL). Table 3a summarizes results for $O_{PM_{10}}$, and Table 3b for $O_{coarse}$. Rankings are by performance of R coefficient for each site.

(a)

| Model | AM (R) | AM_RMSE (µg/m³) | AM_MB (µg/m³) | | BS (R) | BS_RMSE (µg/m³) | BS_MB (µg/m³) | | FKL (R) | FKL_RMSE (µg/m³) | FKL_MB (µg/m³) |
|---|---|---|---|---|---|---|---|---|---|---|---|
| NASA_GEOS | 0.65 | 79.13 | 24.7 | NOA_WRF | 0.61 | 215.47 | 128.66 | NOA_WRF | 0.68 | 64.9 | 40.68 |
| NOA_WRF | 0.51 | 79.25 | 42.39 | NASA_GEOS | 0.59 | 212.6 | 75.92 | DREAMABOL | 0.62 | 44.94 | −7.56 |
| MMM | 0.32 | 53.71 | −14.36 | DREAM8b_V2 | 0.51 | 165.96 | −1.05 | NASA_GEOS | 0.56 | 106.02 | 43.66 |
| MACC_ECMWF | 0.26 | 60.44 | −20.19 | MMM | 0.51 | 167.23 | 7.1 | MMM | 0.56 | 44.88 | −12.89 |
| NCEP | 0.25 | 64.43 | 11.39 | NMMB | 0.5 | 211.68 | −37.31 | DREAM8b_V2 | 0.54 | 51.72 | −7.83 |
| DREAM8_MACC | 0.22 | 62.12 | −29.31 | DREAM8_MACC | 0.47 | 178.55 | −13.81 | DREAM8_MACC | 0.48 | 51.53 | −27.73 |
| DREAM8b_V2 | 0.22 | 60.51 | −17.55 | NCEP | 0.46 | 167.43 | −10.07 | NCEP | 0.46 | 70.06 | 31.63 |
| SILAM | 0.2 | 191.44 | 28.18 | MACC_ECMWF | 0.42 | 186.89 | −75.36 | MACC_ECMWF | 0.43 | 69.81 | −17.38 |
| NMMB | 0.19 | 65.52 | −28.27 | DREAMABOL | 0.41 | 186.52 | −22.64 | NMMB | 0.28 | 63.45 | −33.26 |
| DREAMABOL | 0.17 | 56.11 | −4.15 | EMA_REG4 | 0.28 | 325.89 | 5.96 | EMA_REG4 | 0.24 | 50.77 | −29.62 |
| LOTOS-EUROS | 0.15 | 81.04 | −7.20 | SILAM | 0.26 | 189.39 | 32.42 | SILAM | 0.19 | 72.42 | 16.3 |
| EMA_REG4 | −0.07 | 69.43 | −48.03 | LOTOS-EUROS | 0.08 | 258.6 | 27.86 | LOTOS-EUROS | – | – | – |

(b)

| Model | AM (R) | AM_RMSE (µg/m³) | AM_MB (µg/m³) | | BS (R) | BS_RMSE (µg/m³) | BS_MB (µg/m³) | | FKL (R) | FKL_RMSE (µg/m³) | FKL_MB (µg/m³) |
|---|---|---|---|---|---|---|---|---|---|---|---|
| NASA_GEOS | 0.64 | 89.58 | 44.48 | NOA_WRF | 0.64 | 237.19 | 158.6 | | | | |
| NOA_WRF | 0.52 | 89.31 | 60.03 | NASA_GEOS | 0.52 | 241.7 | 118.21 | | | | |
| MMM | 0.34 | 42.89 | 5.29 | MMM | 0.47 | 166.29 | 47.51 | | | | |
| MACC_ECMWF | 0.26 | 49.57 | −0.43 | DREAM8b_V2 | 0.44 | 162.37 | 39.91 | | | | |
| NCEP | 0.26 | 64.19 | 31.01 | DREAM8_MACC | 0.43 | 172.53 | 26.37 | | | | |
| DREAM8_MACC | 0.23 | 44.95 | −9.77 | NMMB | 0.43 | 216.35 | 2.08 | | | | |
| DREAM8b_V2 | 0.22 | 50.04 | 1.65 | NCEP | 0.39 | 157.88 | 29.14 | | | | |
| NMMB | 0.2 | 52.03 | −8.72 | DREAMABOL | 0.37 | 161.09 | 14.82 | | | | |
| SILAM | 0.19 | 195.98 | 45.78 | MACC_ECMWF | 0.37 | 159.61 | −35.71 | | | | |
| DREAMABOL | 0.18 | 51.11 | 14.04 | EMA_REG4 | 0.25 | 333.95 | 43.98 | | | | |
| LOTOS-EUROS | 0.18 | 77.39 | 11.13 | SILAM | 0.23 | 198.82 | 58.12 | | | | |
| EMA_REG4 | −0.07 | 50.5 | −29.63 | LOTOS-EUROS | 0.1 | 253.05 | 50.9 | | | | |

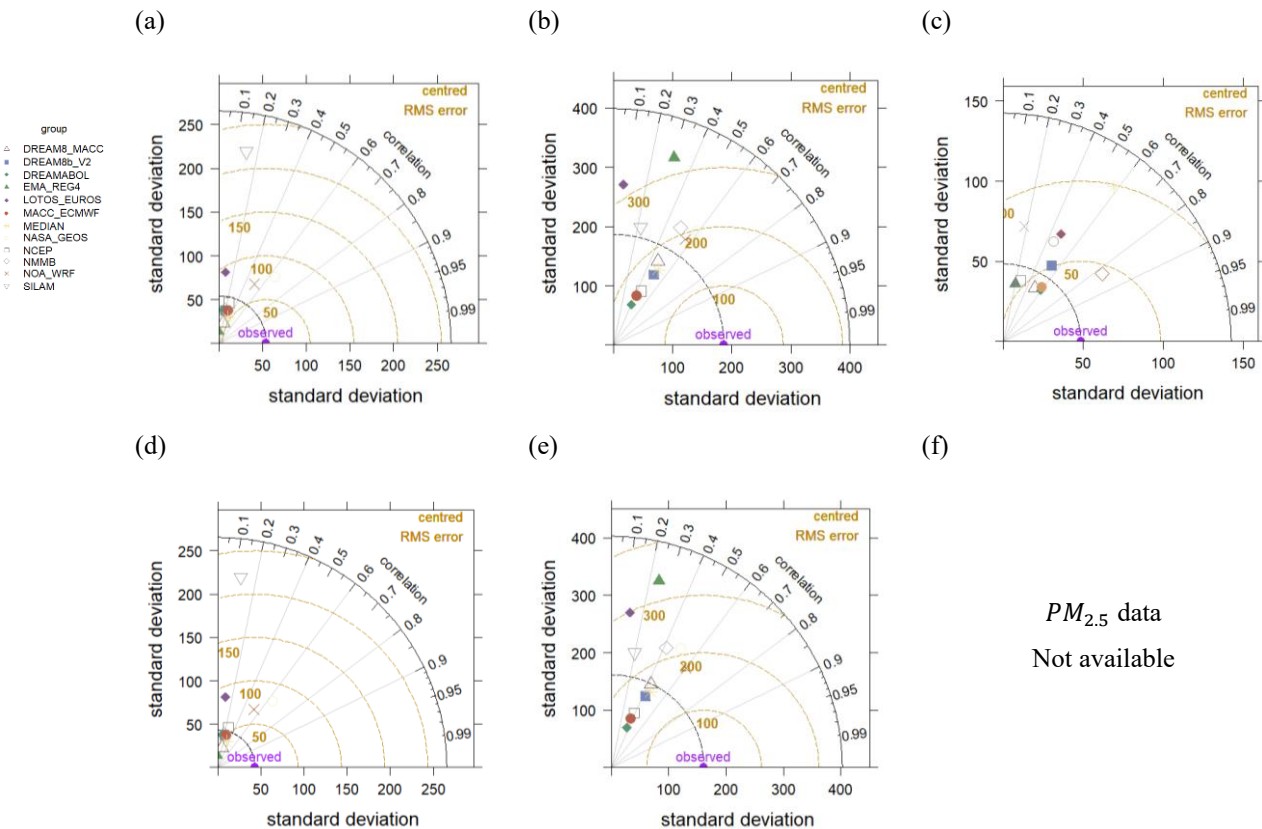

Figure 7: Taylor diagrams for daily PM$_{10}$ during high-dust conditions at the three monitoring sites. Panels show model performance against observations at (a and d) Ayia Marina (AM), (b and e) Be'er Sheva (BS), and (c) Finokalia (FKL). Panels (a–c) correspond to method (i) $M_{dust_{conc}}$ vs $O_{PM_{10}}$ and (d–e) correspond to method (ii) $M_{dust_{conc}}$ vs $O_{Coarse}$. The azimuth indicates the correlation coefficient (R), the radial distance the standard deviation (SD), and the concentric arcs the centered RMSE. Symbols denote the 11 operational models; the multi-model median (MMM) is highlighted. The observational reference is located at R=1 and SD=1.

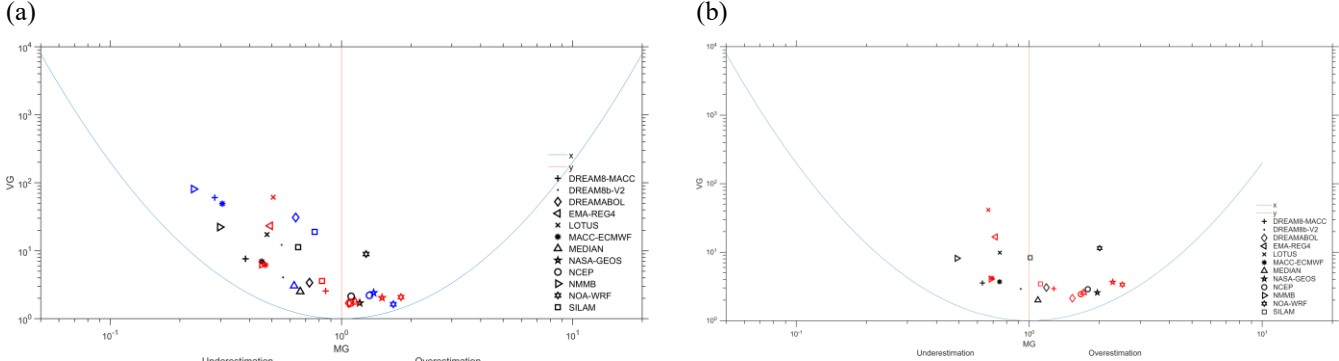

Figure 8: BOOT statistical plots comparing modelled and observed PM$_{10}$ concentrations for the Achilleos et al. (2019) approach at Ayia Marina (AM), Be'er Sheva (BS), and Finokalia (FKL). a) $M_{dust_{conc}}$ vs $O_{PM_{10}}$ and b) $M_{dust_{conc}}$ vs $O_{Coarse}$. Models positioned closer to the origin show better performance. Extreme outliers are excluded for readability.

## 4.2. Categorical statistics

The following results present contingency table analyses in the form of daily heatmaps, illustrating the agreement between modelled and observed PM concentrations at each site. For each dataset and approach, model performance is categorized into three outcomes: overestimation, underestimation, and accurate prediction (hits, defined as model values within ±1 standard deviation of

ground observations). Two modelling approaches are evaluated, $O_{coarse}$ and $O_{PM_{10}}$, to assess how effectively each model captures different particulate matter fractions and their day-to-day variability across sites.

### 4.2.1. Ayia Marina (AM), Cyprus

Figure 9: Heatmaps of modelled dust concentrations ($M_{dust_{conc}}$) against (a) observed coarse fraction ($O_{coarse}$) and (b) observed PM10 ($O_{PM_{10}}$) for the entire study period. Color intensity shows the density of paired values, indicating the level of agreement between modeled dust and observed particulate concentrations.

Figure 9a and b show the heatmaps for the Ayia Marina site, comparing model performance for $O_{coarse}$ and $O_{PM_{10}}$ over the entire study period. All models achieve over 80% hits for both approaches, demonstrating their ability to capture the overall dust conditions at this site. The DREAM8-MACC model stands out with the highest hit rat`e at 97%, indicating excellent performance in predicting both coarse and total PM concentrations. On the other hand, NOA-WRF records the lowest hit rate (81%), reflecting a tendency to overestimate the dust levels in this region, as confirmed by the model's consistent overestimation across multiple metrics in prior analyses. Underestimation remains minimal for Ayia Marina in both approaches, indicating that most models are capturing or over-predicting dust events rather than missing them. This is critical for public health warnings, as underestimation would lead to insufficient precautionary measures.

For the 95th percentile of ground observations (Figure10a and b), AM shows mixed results between hits and overestimations in the $O_{coarse}$ approach. NOA-WRF and NASA-GEOS display the highest overestimations, consistent with their tendency to over-predict extreme dust events, which has been observed across multiple metrics. EMA-REG4, on the other hand, records the fewest overestimations and is more prone to underestimating, continuing its underperformance noted in earlier sections. The overall hit percentages remain high for most models, but overestimations dominate during peak dust events. In the $O_{PM_{10}}$ approach, the results are similar, with NOA-WRF and NASA-GEOS again leading in overestimations, but no strong trends emerge beyond these individual cases. EMA-REG4 again shows high underestimation, making it the weakest performer during these peak dust events.

For the specific dates examined, following the third approach of Subsection 3.3, the $O_{coarse}$ approach shows (Figure10a) that most models perform well, capturing ground concentrations within one standard deviation. EMA-REG4 has the highest hit percentage, which is notable given its underperformance in other scenarios. DREAM8-MACC also performs well, while NOA-WRF shows the highest overestimation rate, continuing the trend seen in previous sections. Underestimations remain low, indicating that most models effectively predict peak dust levels.

### 4.2.2. Be'er Sheva (BS), Israel

The performance at Be'er Sheva, shown in Figure9a and b, is more varied compared to AM. For the $O_{coarse}$ approach, the models exhibit a broader range of results. MACC-ECMWF and NMMB models show the highest hit percentages, indicating strong predictive capabilities for coarse particles. In contrast, DREAM8b-V2 has the lowest hit rate, suggesting a struggle to accurately capture dust levels at this location, particularly for coarse particles. In the $O_{PM_{10}}$ approach, the performance remains largely the same, with MACC-ECMWF and NMMB continuing to outperform other models. However, DREAM8b-V2 again records the lowest hit rate, reinforcing its overall weaker performance at this site, which was also observed in the earlier analyses of correlation and bias metrics. Overestimation is more frequent than underestimation, particularly for NOA-WRF, which tends to significantly overestimate PM concentrations, leading to potential overreactions in dust event management.

For the 95th percentile of ground observations, overestimations dominate the results for both approaches ($O_{coarse}$ and $O_{PM_{10}}$), as depicted in Figure10a and b. This site's proximity to major dust sources in the Middle East may contribute to the over-prediction, as models struggle to account for rapid dust inflow. NOA-WRF shows the highest overestimation, with no underestimations

recorded for the $O_{coarse}$ approach, reinforcing the model's consistent bias towards overestimating dust levels at this site. EMA-REG4, which previously showed underestimations at other sites, also demonstrates high underestimations for $O_{PM_{10}}$, indicating that it struggles to capture the dust dynamics in Be'er Sheva.

For the specific dates examined, at Be'er Sheva, overestimations dominate again in the $O_{coarse}$ approach, with NOA-WRF and NASA-GEOS showing the highest overestimation percentages. EMA-REG4, typically prone to underestimations, continues to show a relatively balanced performance between hits and underestimations. In the $O_{PM_{10}}$ approach (Figure 11b), the results are consistent, with high overestimation rates, reinforcing the models' struggle to accurately predict dust levels in this challenging region.

### 4.2.3. Finokalia (FKL), Greece

At Finokalia, model performance is consistently strong, with all systems achieving hit percentages above 88% (Figure9). The LOTOS-EUROS model performs best, reaching a 100% hit rate for $O_{coarse}$, indicating that it accurately captures dust conditions at this site. This high performance contrasts with LOTOS-EUROS's weaker results at other sites, like Be'er Sheva, highlighting its region-specific strengths. Conversely, NCEP records the lowest hit percentage, suggesting challenges in predicting both coarse and total PM at Finokalia. Finokalia, with its more consistent results, supports the idea that this site, being less influenced by anthropogenic sources, allows models to focus more on transboundary dust transport, leading to higher accuracy across most models. This observation aligns with the results from previous sections, where high correlations and lower biases were recorded for many models at this site.

For this area, there are no trends using the 95[th] percentile of ground observations, in either approach, with most models showing a balance between hits, overestimations, and underestimations (Figure10). EMA-REG4 and NMMB record the highest underestimations, while NOA-WRF shows the highest overestimations, consistent with its overall behaviour across the sites. The hit percentages remain relatively high, but the variability indicates that extreme dust events in Finokalia are more challenging for models to predict accurately, likely due to the site's geographical characteristics.

For the specific-dates approach, using $O_{PM_{10}}$ data, (Figure11b), most models exceed 50% hit rates, with DREAMABOL performing best. NOA-WRF remains the only model to exceed 80% overestimation, reinforcing its bias toward higher PM$_{10}$ levels during dust events. Overall, Finokalia exhibits robust model agreement, but continued overestimation of peak magnitudes underscores the need for improved representation of intense long-range dust transport episodes.

(a)                                                                 (b)

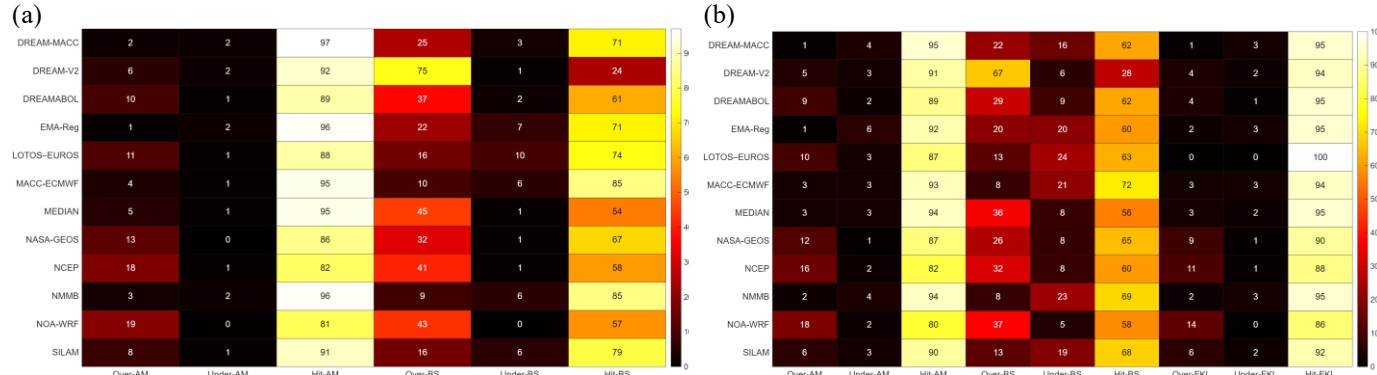

Figure 9: Heatmaps of modelled dust concentrations ($M_{dust_{conc}}$) against (a) observed coarse fraction ($O_{coarse}$) and (b) observed PM$_{10}$ ($O_{PM_{10}}$) for the entire study period. Color intensity shows the density of paired values, indicating the level of agreement between modeled dust and observed particulate concentrations.

(a)                                                                 (b)

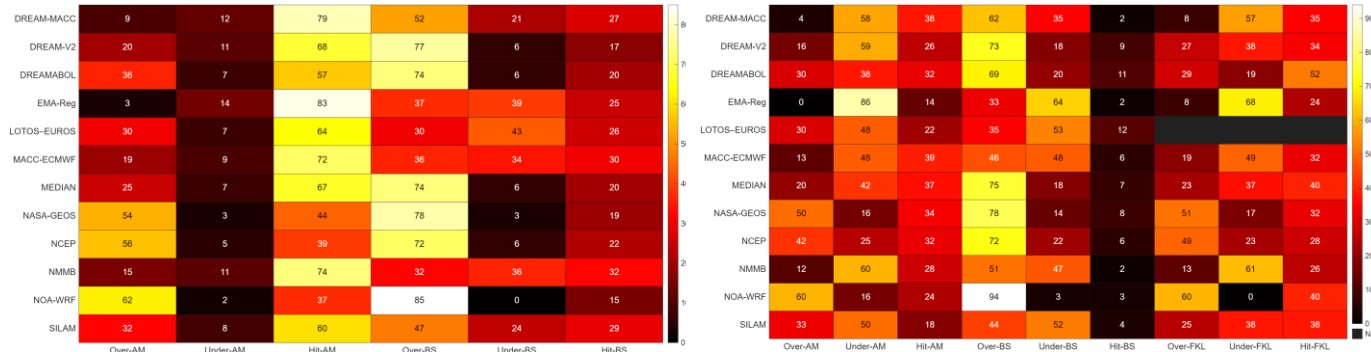

Figure 10: Heatmaps of modelled dust concentrations ($M_{dust_{conc}}$) against (a) observed coarse fraction ($O_{coarse}$) and (b) observed PM10 ($O_{PM_{10}}$) for the 95th percentile of observed data. Color intensity indicates the density of paired values, highlighting model–observation agreement during extreme dust events

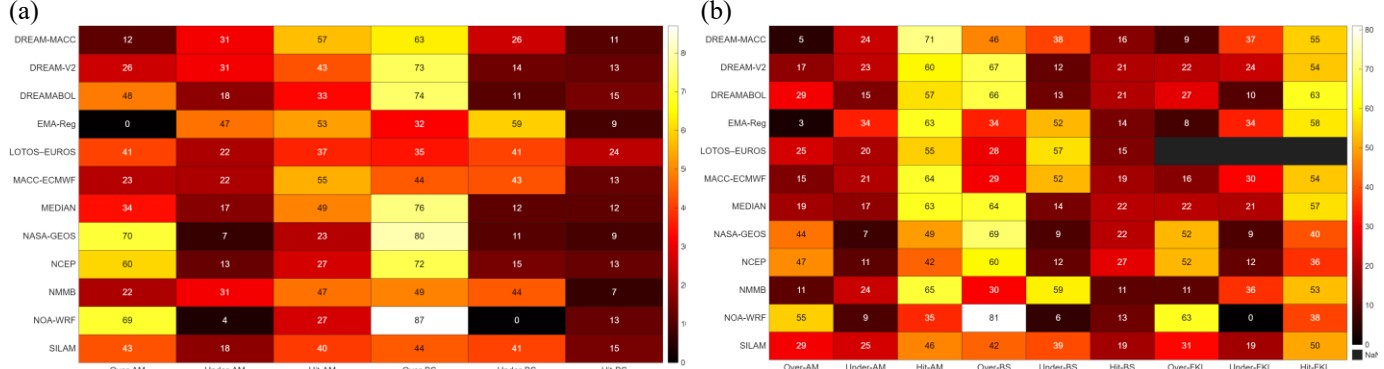

Figure 11: Heatmaps of modelled dust concentrations ($M_{dust_{conc}}$) against (a) observed coarse fraction ($O_{coarse}$) and (b) observed PM10 ($O_{PM_{10}}$) for dust days (Achilleos et al., 2019). Color intensity shows the frequency of paired values, illustrating model performance under dust-dominated conditions.

Overall, no model demonstrates exceptional accuracy across all sites, approaches and datasets. A high hit rate of models accurately predicting observations is observed only in the first dataset, which assesses performance over the entire study period. Interestingly, models like EMA-REG4 and NMMB, which consistently perform poorly in other evaluation methods, show significantly better results when evaluated using contingency table methods.

## 5.    Evaluation of Configuration Settings in the performance of Operational Dust Forecasting Models

Differences in model configuration lead to measurable variations in performance across EMR sites. Key configuration elements shaping model performance include the representation of the near-surface vertical structure (including first-layer height), the use of aerosol and meteorological data assimilation and the horizontal resolution; their relative influence varies by site and evaluation subset. Over the full period, systems that combine fine near-surface vertical structure with data assimilation, most notably NASA-GEOS, achieve higher correlations and balanced errors across sites (e.g., R = 0.71 at AM, 0.65 at BS, and 0.64 at FKL for PM₁₀), with |MB| ≤ 15 μg/m³ and RMSE typically ≤ 75 μg/m³ (Table 3a–b; Figures 3–4). The MMM similarly exhibits near-leading, stable performance by reducing the influence of outliers at each timestamp and delivering top-rank RMSE with moderate-to-high correlations.

The influence of vertical structure is most evident in the representation of surface mixing and diurnal gradients. Finer vertical discretization and lower first-layer heights consistently improve correlation and reduce RMSE across sites. Conversely, coarser vertical structures under-capture near-surface variability at FKL and tend to amplify positive bias at BS, shifting MB away from

zero. These effects are substantiated in the updated Table 2, which reports first-layer heights (as provided by modeling teams), clarifying cross-system differences that align with the observed rankings in Table 3.

Horizontal resolution primarily affects the timing and amplitude of intense dust episodes rather than average performance across the full period. NOA-WRF illustrates this pattern: it achieves strong event-level performance (e.g., R = 0.91 at FKL in the 95th percentile subset) and maintains competitive full-period skill at BS (R = 0.62). However, it also overestimates near-source concentrations, with MB ≈ +10 to +40 μg/m³ and RMSE in the 70–110 μg/m³ range. At sites where long-range transport dominates, such as Finokalia and Ayia Marina, RMSE values are lower (≈ 20–40 μg/m³ at FKL and 25–35 μg/m³ at AM), and finer grid resolution provides limited additional benefit once transport phasing is adequately captured.

The application of data assimilation systematically improves transport phasing and magnitude, raising correlations and reducing biases across both full-period and event-focused subsets. NASA-GEOS, which assimilates relevant meteorological and aerosol fields, consistently maintains R ≥ 0.60 across all sites and subsets, with MB near zero and RMSE ≤ 75 μg/m³. Other data assimilation -enabled models (e.g., MACC-ECMWF, NCEP) show ranking gains under confirmed dust-day conditions relative to the 95th percentile subset, suggesting that multi-parameter event identification enhances the benefit of data assimilation by better aligning vertical structure and timing with observations.

Additional factors such as meteorological forcing type (e.g., analysis vs forecast) and source emission parameterizations further influence site-specific skill. At BS, close to major dust sources, models exhibit the largest variance and error spread (MB up to +37 μg/m³; RMSE > 100 μg/m³ in several systems), reflecting sensitivity to uplift intensity and boundary inflow. In contrast, AM and FKL (representing regional background conditions) display more systematic underestimation (MB ≈ −5 to −15 μg/m³) and lower RMSE values (25–35 μg/m³ at AM; 20–40 μg/m³ at FKL), consistent with reduced influence from local sources and more predictable transport pathways.

Within this context, the MMM provides robust central performance by computing the median across harmonized model outputs at each timestamp and comparing the resulting time series to observations. Over the full period, MMM achieves R ≥ 0.60 at one or more sites, with low RMSE and |MB| ≤ 15 μg/m³. Substitution of $O_{coarse}$ for PM₁₀ generally reduces mean biases without significantly altering amplitude errors: at AM, MMM MB improves from −9.84 to +0.66 μg/m³, and DREAM8-MACC from −13.87 to −3.43 μg/m³, while RMSE remains within the range of ~20–80 μg/m³. This suggests that coarse-fraction filtering corrects offset but does not resolve the full amplitude spread during dust events.

In summary, the most influential configuration elements are:

(i) vertical resolution near the surface, including first-layer height, and

(ii) the presence of data assimilation.

Horizontal resolution contributes less to full-period averages but enhances event timing and intensity during extremes. These relationships explain the consistent top-tier performance of NASA-GEOS, the stability of the MMM, and the event-specific strengths (and biases) of NOA-WRF. From an operational perspective, MMM-guided forecasts, supported by data assimilation-enabled systems with fine vertical resolution and routine evaluation against both $O_{PM_{10}}$ and $O_{coarse}$, offer a practical strategy to reduce bias, limit error growth during dust intrusions, and enhance day-to-day forecast reliability across the EMR.

## 6.     Conclusions and Future Directions in Dust Modelling in EMR

This study provides a comprehensive evaluation of eleven operational dust models and a multi-model median (MEDIAN) in forecasting PM concentrations across the EMR. Evaluating operational dust forecasting numerical models against ground measurements is essential to identifying for determining which models perform better in different areas. Such evaluations are crucial for enhancing the accuracy and reliability of DDS - EWS, as these systems rely on robust forecasting models to provide

timely alerts for hazardous dust events. Despite their importance, few studies have conducted such evaluations against ground-level measurements, highlighting a critical gap that this research aims to address.

## 6.1. Key findings

The results clearly indicate that certain models outperform others in predicting ground-level PM concentrations across specific regions, although their performance varies based on the evaluation method applied. Notably, no single model consistently achieves accurate predictions across all three regions, underscoring the need for model-specific adaptations or improvements tailored to regional characteristics. The key findings of this study are summarised as follows:

a) **Model Performance Across Sites**: The accuracy of individual models varied significantly across the three sites (Ayia Marina, Finokalia, and Be'er Sheva). NASA-GEOS emerged as the most consistent performer, particularly at Ayia Marina and Be'er Sheva, benefiting from a high-resolution vertical structure, data assimilation, and robust meteorological inputs via the GEOS-5 driver. Conversely, models like EMA-REG4 showed substantial limitations, especially in accurately predicting PM levels in Cyprus, indicating that regional drivers and lower-resolution configurations may not be suitable for cross-boundary events in the EMR.

b) **Enhancing Early Warning Systems:** The findings of this study can provide valuable insights for improving EWS and mitigation strategies in the EMR. By identifying the strengths and weaknesses of the evaluated dust models, particularly in predicting extreme events and PM concentrations, these results can help tailor EWS to the region's specific needs. For example, models such as NASA-GEOS and NOA-WRF, which demonstrated higher accuracy and correlation during intense dust events, can be prioritized for operational forecasting to provide timely warnings to vulnerable populations. Additionally, the multi-model median (MEDIAN), with its reduced uncertainty, can enhance the reliability of forecasts, supporting decision-makers in public health protection and emergency preparedness. These improved predictions are essential for issuing health advisories, minimizing exposure to hazardous PM levels, and informing policy measures to mitigate the adverse impacts of DDS events on air quality and public health.

c) **Differences between 95$^{th}$ percentile dataset and dust days by Achilleos et al.2019:** The results from these 2 datasets especially for the best performing models of NASA GEOS and NOA WRF or the least performing such as EMA REG, are very similar as both datasets are set to describe the highest dust events. Even though they seem similar after the best and least performing models, the rankings vary as the other models are not as consistent as the ones mentioned before. For example, MACC ECMWF and NCEP models have better ranking when the confirmed dust days is used indicating that the multi parameter criteria approach in identifying dust days helps models to improve in their forecasts. Also, the dust days approach lowers the range between values such as RMSE and MB as shown in Tables 4 and 5 and in the BOOT methodology graphs in Figures 6 and 8 for both approaches respectively, thus decreasing the uncertainty between approaches.

d) **Impact of Model Configurations on Performance:**

    i. **Meteorological Drivers and Data Assimilation**: ECMWF and GEOS-5-driven models demonstrated higher correlations with observed data, particularly when data assimilation was included. Data assimilation improves models' responsiveness to rapid dust influx and significantly enhances reliability in high-dust scenarios, underscoring its necessity in complex regions like the EMR. Optimizing dust forecasting models is crucial for improving predictive accuracy, especially in regions affected by transboundary dust transport. These advancements align with the goals of initiatives like the Horizon Europe CiROCCO project, which aims to strengthen dust storm monitoring through integrated predictive frameworks.

    ii. **Horizontal and Vertical Resolution:** Models with finer horizontal and vertical resolutions, such as NOA-WRF (0.19º x 0.22º, 30 layers) and NASA-GEOS (0.25º x 0.3125º, 72 layers), generally showed improved spatial and temporal accuracy

across sites. Enhanced vertical resolution, particularly for capturing dust plumes at altitude, contributed to better event tracking, while horizontal detail supported finer spatial representation critical for sites like Be'er Sheva with localized dust sources.

       iii.      **First Layer Height and Emission Schemes:** Lower first-layer heights (10-20 m) improved the models' capacity to capture ground-level concentrations accurately, aligning predictions with near-surface PM10 observations. Emission schemes, especially those incorporating sandblasting (e.g., Marticorena and Bergametti, 1995), supported accurate predictions during intense dust events by modelling larger particle transport effectively.

    **e)**    **Role of Multi-Model Ensemble (MEDIAN):** The multi-model median, which averaged predictions across all models, consistently reduced biases and RMSE values, supporting its effectiveness as a robust forecasting operational approach. Although it does not consistently outperform individual models such as NASA-GEOS, the MEDIAN's reduced error scores make it a reliable option for broader operational applications in the Eastern Mediterranean region (EMR).

    **f)**    **Approaches and methods of evaluation:** The different approaches $O_{PM_{10}}$ and $O_{coarse}$, different datasets and methods of evaluation have shown that some model's performance can be perceived differently if not properly evaluated by different methods. An example is EMA-REG4, that while being evaluated by performance metrics and BOOT graph methodology it has a very poor performance but on the contingency tables it has one of the best overall scores. Furthermore, swapping from $O_{PM_{10}}$ to $O_{coarse}$, it shows how sensitive are models to fine materials as they do or do not take them into account changing an underestimation to overestimation and vice verca, while their correlation is mostly decreasing.

This study has limitations, including the absence of recent PM2.5 data in some regions and potential changes in model configurations that might affect performance. Despite these limitations, operational models remain reliable for predicting DDS events and provide valuable mitigation insights (Eleftheriou et al., 2023). However, continuous technological advancements are essential to enhance dust prediction capabilities, particularly concerning the duration and intensity of DDS events.

## 6.2. Future Directions

To advance dust forecasting in the Eastern Mediterranean Region (EMR), targeted model improvements and data integration strategies are essential. Enhancing data assimilation through high-frequency satellite data, such as MODIS AOD, and incorporating localized PM monitoring can improve real-time prediction accuracy, particularly when dynamic field measurements like PM2.5 data are included. Importantly, expanding data assimilation to include more in-situ measurements by strengthening monitoring networks, especially in underrepresented source areas, would enhance the accuracy of model predictions and ensure better representation of dust transport from these critical regions. Also, sometimes the models fail producing negative or even zero concentration thus influencing the evaluation technique used to assess their performance. Such results should be redefined to a Limit of Detection (LOD) consistent with observation techniques to provide a more realistic result and the result to be valid in order to be evaluated.

The variability in prediction accuracy across models also highlights the need for improved, site-specific emission parameterizations. Adaptive emission schemes that account for local soil characteristics, vegetation cover, and land use would be especially beneficial for urban-adjacent and desert-proximal sites like Be'er Sheva. Furthermore, results consistently favouring models with finer spatial (0.1°–0.3°) and vertical resolutions (over 50 layers) suggest that these refinements are crucial for accurately capturing dust dynamics and transboundary events impacting EMR air quality.

Customizing multi-model median strategies by including only the top-performing models for each site could enhance forecast precision while preserving the bias reduction advantages seen in multi-model medians. Such an approach leverages specific model strengths under varied conditions, especially during peak dust events. Expanding model validation efforts to include diverse datasets—such as vertical aerosol profiles, aerosol chemistry, and source-specific particle data—would further refine model

parameterizations and improve reliability. Collaborating with regional networks to utilize observational data from North Africa, the Middle East, and EMR could substantially strengthen model robustness.

Future efforts should focus on refining model parameterizations, integrating localized field data, and validating models against diverse datasets to ensure predictive robustness and reliability. Overall, the results underscore the region-specific nature of dust forecasting and highlight model configurations and features that enhance accuracy at individual sites.

**Code and Data Availability.**

Model forecast data from the SDS-WAS Barcelona Dust Forecast Center ([https://sds-was.aemet.es/](https://sds-was.aemet.es/)) are openly accessible and were archived during the analysis period. The ground-based measurement data used in this study are available upon request from the authors. This study was performed using custom scripts in MATLAB (more details can be found at https://www.mathworks.com) and R (it is available at https://www.r-project.org). While no new modelling code was developed in this study, the scripts used for data processing, correlation analysis, and performance evaluation are available upon request from the corresponding author.

**Author contributions.**

MNKA, PM, and AE contributed to manuscript preparation. MNKA and PY provided the financial support for this study. MNKA also contributed to the investigation, provided resources, and participated in the supervision of the study. PM had the conceptualization, investigation, formal analysis, and supervision of this study. AE and PM performed the formal analysis of the data; AE also prepared the original draft of the manuscript. NK, IK, EV, and CS were responsible for data curation and contributed to manuscript review and editing. PK and PY contributed to writing – review and editing. All co-authors reviewed and approved the final version of the manuscript.

**Competing interests.**

The contact author has declared that none of the authors has any competing interests.

**Acknowledgments**

This research work is part of the CiROCCO Project, funded by the European Union (Grant no. 101086497). Views and opinions expressed are, however, those of the author(s) only and do not necessarily reflect those of the European Union or REA. Neither the European Union nor the granting authority can be held responsible for them. Moreover, the authors are grateful for the financial support from the LIFE+-MEDEA Program under Grant Agreement LIFE16 CCA/CY/000041. The authors would also like to express their deepest gratitude to the data providers, the organizations, and their researchers who develop and operate the operational models, as well as to the SDS-WAS portal for the collection and availability of the data.

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
