# Peer review of "How accurate are operational dust models in predicting Particulate Matter (PM) levels in the Eastern Mediterranean Region? Insights from PM Surface Concentrations"

_EGUsphere, 2025_

## Referee Comment (RC1)

Review of egusphere-2025-2739 entitled "How accurate are operational dust models in predicting Particulate Matter (PM) levels in the Eastern Mediterranean Region? Insights from PM Surface Concentrations" by Andreas Eleftheriou et al.

**General**

The manuscript entitled "How accurate are operational dust models in predicting Particulate Matter (PM) levels in the Eastern Mediterranean Region? Insights from PM Surface Concentrations" by Andreas Eleftheriou et al. provides an assessment of the performance of eleven operational dust forecasting models and a multi-model ensemble through comparisons against surface PM measurements at three sites in the Eastern Mediterranean Region (EMR), Ayia Marina (AM) in Cyprus, Be'er Sheva (BS) in Israel and Finokalia (FKL) in Crete. The evaluation is done using specific established statistical metrics, namely correlation coefficient, R, Mean Bias, MB, and Root Mean Square Error, RMSE. The obtained results reveal a substantial variability in the models' accuracy that no single model consistently achieves accurate predictions across all three regions and in all conditions (entire study period and days with high dust loadings).

The manuscript adds to the scientific community's knowledge about the performance of operational dust models. Nowadays, a significant effort is made on developing and implementing such models to monitor dust levels in the atmosphere, and also warning the public about hazardous dust episodes, at regional or global scales. Given that these models differ between them in their spatial resolution, meteorological drivers, emission schemes or data assimilation procedures, it is important to intercompare them and to draw conclusions on which model(s) outperform. In this meaning, the study is interesting, although the conclusion drawn is not clear as to which model does so, in overall. While this is a bit disappointing, the study proves and convinces the reader that this happens given the multi-parametric problem, in the sense that many and combined factors play role and determine the overall model performance. While some questions remain unanswered (as explained below), it is more or less understood that a model can perform well at some site, while not in another, or better/worse under different dust loading conditions.

Based on the above, and the fact that the analysis is correct and complete at a significant level, while the text is well organized and written, I recommend publication of the manuscript subject to some corrections and recommendations for revision suggested below.

*Main Comments*

1. Further reference to the existing literature dealing with Mediterranean dust storms/episodes should be made. In particular, additional references should be made to papers dealing with increasing/decreasing desert dust storms in the Mediterranean Basin.

2. The style of discussion of the results obtained is a bit tiring for the reader. For example, discussing the results station by station in section 4.1 may get the reader

tired and lost in the details given. What matters is the comparative analysis, which is discussed in the last paragraph (section 4.1.1). This kind of discussion can be expanded referring to Figures 3 and 4. Also, Table 3 can be restructured to show rankings of the 11 models performance regarding the 3 different stations and the statistical metrics.

3.  Improve and make more descriptive the captions of some Figures (e.g. Figures 9, 10, 11) to help the reader to more easily and rapidly understand what is shown.

4.  The style of discussing results in section 5 should be improved by: (i) reporting the values of statistical metrics wherever recalled/reported in the text, (ii) specifying if the statements made with reference to the models' performance concern the entire study period or other conditions, e.g. the 95[th] percentile pf observed PM concentrations.

5.  Provide arguments to infer responses to questions as to why a model, e.g. the NASA-GEOS, outperforms at AM and BS but not in FKL, or why a model, e.g. NOA-WRF outperforms during intense dust events, but not the same in overall.

*Minor comments*

1. Line 64: Yet, the satellite detection algorithms have progressed with time, including geostationary satellites as well (e.g. Kolios and Hatzianastassiou, 2019) exempt from the limitations of the polar orbiting satellite-based algorithms.

2. Line 93: the reference "Varga et al., 2014)" misses in the list of references.

3. Section 2: Dust storms in the MB and their spatiotemporal characteristics are better captured by satellites (e.g. Gkikas et al., 2013, 2016).

4. Line 110: The range of measured/exported particles' size should be given (it is necessary information to be used in the comparison with the corresponding sizes of the 11 models) since differences with models can partly explain the PM overestimations/underestimations by the models.

5. Line 131: replace "develop …" by "developed …".

6. Lines 133: It would be useful to indicate these dust storm days on Fig. 2, probably using different symbols for each station. Thus, readers can have an idea about the intensity of these dust storms at every site. Also, are there common days in the 3 stations out of the reported 106, 88 and 101 dust storm days?

7. Table 1: What kind of radiation interactions are those reported in this Table? Aerosol-radiation interactions or others as well, e.g. aerosol-cloud? Please specify.

8. Table 2: at what height is the first level above surface for the NASA_GEOS and NOA_WRF models, why are the corresponding values missing in the Table?

9. Line 167: use a parenthesis after the sum symbol, i.e. put the difference "$M_i-O_i$" in a parenthesis.

10. Line 179: relative bias and relative RMSE would be equally interesting metrics to show (as bias and RMSE).

11. Figure 3: Change "MEDIAN" to "MMM" for consistency with the rest of paper. Do the same in Figures 5 and 7.

12. Line 310: Figures 3 and 4 are not discussed and mentioned in the text, reference to them should be made.

13. Line 330: does "… R=0.62 …" should read "… R=0.55 …"?

14. Discussion in section 4.1.2: It is also interesting to discuss if models perform better or worse for high-dust events compared to all cases, also providing possible explanations for the improvement/deterioration. Some models, e.g. NOA_WRF, perform better for high-dust events in some locations (in FKL in this case) while doing worse in other locations (BS and AM), why does this happen?

15. End of section 4.1: A general assessment should made referring to whether the findings for the identified dust days by Achilleos et al. (2020) methodology are similar to those drawn from the high-dust days analysis of the previous section or not. It is essential to see if the methodology applied to identify dust events affects the results referring to the comparative model's performance analysis (keeping in mind that basically they should not do so).

16. Discussion of Figures 9, 10 and 11: Make a comment on existing differences between Figures 10 and 11 since the nature of the results shown on these figures is about similar (as they both refer to days with high dust loadings).

17. Lines 502-505: This is a (probably the most) typical statement reflecting the complexity of the problem addressed, concerning the distinction of which is/are the model/models that perform better than others, overall. In spite of the differences existing between the models at various levels, they are all performing more or less similarly, perplexing the situation/problem.

18. Lines 508-510: Remove the empty line and link the text (from "and" to "Table 2").

19. Section 5: The results show that different factors, e.g. vertical/horizontal resolution, height of first model level etc. are used as criteria for evaluating the performance of the models. Yet, it seems that some models are superior to others with respect to a specific factor, while other model(s) are superior with respect to other factor(s). Thus, it seems that a kind of counterbalance exists, leading to a roughly similar performance of the models. Would it be possible to draw, based on the overall performance of the models, a conclusion about which factor/factors is/are the most important for the model's performance?

20. Lines 591-595: What is reported here is somewhat worrying. Why/how the performance of a model should change depending on the selected approach/method for the evaluation?

**References**

Gkikas, A., Hatzianastassiou, N., Mihalopoulos, N., Katsoulis, V., Kazadzis, S., Pey, J., Querol, X., and Torres, O.: The regime of intense desert dust episodes in the Mediterranean based on contemporary satellite observations and ground

measurements, Atmos. Chem. Phys., 13, 12135–12154, https://doi.org/10.5194/acp-13-12135-2013, 2013

Gkikas, A., Basart, S., Hatzianastassiou, N., Marinou, E., Amiridis, V., Kazadzis, S., Pey, J., Querol, X., Jorba, O., Gassó, S., and Baldasano, J. M.: Mediterranean intense desert dust outbreaks and their vertical structure based on remote sensing data, Atmos. Chem. Phys., 16, 8609–8642, https://doi.org/10.5194/acp-16-8609-2016, 2016

Kolios, S.; Hatzianastassiou, N. Quantitative Aerosol Optical Depth Detection during Dust Outbreaks from Meteosat Imagery Using an Artificial Neural Network Model. Remote Sens. 2019, 11, 1022. https://doi.org/10.3390/rs11091022

---

## Author Response (AR1)

**_Response to comments from the Editors and Reviewers:_**

We would like to thank the Editor and the Reviewers for their thorough evaluation and constructive feedback on our manuscript. We greatly appreciate the time and effort invested in reviewing our work. The feedback has been invaluable in improving the clarity, rigor, and overall quality of the manuscript. In the revised version, we have carefully addressed all comments provided by the reviewers. Where revisions were suggested, we have incorporated the requested changes and clarified ambiguities. A detailed, point-by-point response follows, outlining how each comment was addressed in the revised manuscript. All changes are clearly marked in the revised version to facilitate re-evaluation. We hope that the revisions meet the expectations of the reviewers and the editorial team, and we respectfully submit the manuscript for further consideration.

**Referee #1:**

We sincerely thank Referee #1 for the constructive and supportive review of our manuscript. We appreciate the recognition of the study's contribution and the thoughtful comments that will help us improve the clarity, structure, and interpretative depth of the manuscript. We fully agree with the reviewer's conclusion that no single model can be considered as consistently well-performing across the Eastern Mediterranean. This is also a key message of our study, as the evaluation shows that model skill depends strongly on site-specific characteristics. In general, NASA-GEOS tends to achieve the best scores, but its performance still varies between locations. Proximity to dust sources and local topographic features clearly influence model skill, which explains why a model may perform well at one site but less so at another, or under different dust loading conditions. We will emphasize this point more clearly in the revised conclusions to avoid any ambiguity.

Below we provide detailed responses to all comments and describe the revisions made.

**MAIN COMMENTS:**

**Comment #1: Further reference to the existing literature dealing with Mediterranean dust storms/episodes should be made. In particular, additional references should be made to papers dealing with increasing/decreasing desert dust storms in the Mediterranean Basin.**

Response: We thank the reviewer for this insightful suggestion. In response, we have expanded the Introduction to include additional literature addressing both advancements in satellite-based dust detection and long-term trends in desert dust storm activity across the Mediterranean Basin.

Specifically, we now cite *Kolios & Hatzianastassiou (2019)* to highlight the role of geostationary satellite platforms in improving temporal coverage and enhancing dust monitoring capabilities compared to traditional polar-orbiting satellites. We also incorporated recent studies documenting regional variability and long-term trends in Mediterranean dust storm frequency and intensity, including *Gkikas et al. (2013, 2016), Salvador et al. (2022)*. These works reveal regionally contrasting behaviours, showing decreasing dust storm activity in the western Mediterranean and stable or slightly increasing trends in the eastern basin. In particular, *Salvador et al. (2022)* reported a statistically significant upward trend in the occurrence and intensity of African Dust Outbreaks (ADOs) over the period 1948–2020, reflecting enhanced air mass transport from North Africa toward the western Mediterranean Basin and an intensification of dust episodes in recent decades.

The revised text in the Introduction now reads as follows (Section: Introduction, Lines 65–73):

"Over time, satellite detection algorithms have significantly improved, incorporating geostationary sensors in addition to traditional polar-orbiting platforms (e.g., Kolios and Hatzianastassiou, 2019). Unlike polar-orbiting satellites, geostationary platforms provide continuous temporal coverage and are not constrained by the same revisit-time limitations, thereby enhancing the monitoring of dust transport and diurnal variability. These approaches, while valuable, are subject to inherent limitations. For example, cloud cover can significantly reduce the accuracy of satellite retrievals, while the low temporal resolution of polar-orbiting satellites constrains their ability to capture rapid dust transport dynamics and event-scale variability (Kazadzis et al., 2009). Nevertheless, ongoing initiatives, such as the Horizon Europe CiROCCO project, are designed to address these limitations by developing integrated predictive frameworks and improving both the spatial and temporal coverage of dust monitoring and forecasting systems. "

In addition, the revised text in the Section 2: "Characterizing DDSs in the EMR: Sources and Monitoring" now reads as follows (Lines 99–109):

"Recent studies employing combined satellite and ground-based observations have provided a more detailed picture of these events. For instance, Gkikas et al. (2016), using an algorithm that integrates multi-sensor satellite data with in-situ measurements, found that strong dust storms occur more frequently throughout the year in the western Mediterranean Basin, whereas the most intense events tend to develop in the central region. The study also identified a clear seasonal pattern: dust storms are more common during summer in the western Mediterranean, while they predominantly occur in spring across the central and eastern parts. Earlier work by Gkikas et al. (2013) similarly demonstrated that the spatiotemporal characteristics of Mediterranean dust storms are more effectively captured through satellite-based retrievals, emphasizing the key role of remote sensing in characterizing regional dust dynamics. In addition, Salvador et al. (2022) reported a statistically significant upward trend in the occurrence of African Dust Outbreaks and their intensity over the period 1948–2020. These findings indicate an increase in the frequency of air mass transport from North Africa toward the western Mediterranean Basin and a corresponding intensification of dust episodes during recent decades. "

**Comment #2: The style of discussion of the results obtained is a bit tiring for the reader. For example, discussing the results station by station in section 4.1 may get the reader tired and lost in the details given. What matters is the comparative analysis, which is discussed in the last paragraph (section 4.1.1). This kind of discussion can be expanded referring to Figures 3 and 4. Also, Table 3 can be restructured to show rankings of the 11 models' performance regarding the 3 different stations and the statistical metrics.**

Response: We fully restructured Section 4.1 to improve readability and scientific synthesis. The previous station-by-station description was replaced by a grouped and comparative approach, in which models were organized according to their overall performance (high, moderate, or low). This structure allowed for a concise narrative emphasizing cross-model and cross-site comparisons rather than repetitive descriptions.

The revised text explicitly referred to Figures 3 and 4 and the new Table 3, which was reformatted into two subtables (for PM₁₀ and $O_{coarse}$) to present ranked statistics and facilitate comparison. These modifications made the section more compact, informative, and easier to follow.

**Comment #3 Improve and make more descriptive the captions of some Figures (e.g. Figures 9, 10, 11) to help the reader to more easily and rapidly understand what is shown.**

Response: We revised all figure captions to ensure that each could be understood independently from the main text. The new captions include clear explanations of the datasets, variables, and metrics displayed, as well as short interpretative notes highlighting the main message of each figure.

Specifically, we have revised the captions as follows:

Figure 1: Heatmaps of modelled dust concentrations ($M_{dust_{conc}}$) against (a) observed coarse fraction ($O_{coarse}$) and (b) observed PM10 ($O_{PM_{10}}$) for the entire study period. Color intensity

shows the density of paired values, indicating the level of agreement between modeled dust and observed particulate concentrations.

Figure 2: Heatmaps of modelled dust concentrations ($M_{dust_{conc}}$) against (a) observed coarse fraction ($O_{coarse}$) and (b) observed PM10 ($O_{PM_{10}}$) for the 95th percentile of observed data. Color intensity indicates the density of paired values, highlighting model–observation agreement during extreme dust events

Figure 11: Heatmaps of modelled dust concentrations ($M_{dust_{conc}}$) against (a) observed coarse fraction ($O_{coarse}$) and (b) observed PM10 ($O_{PM_{10}}$) for dust days (Achilleos et al., 2019). Color intensity shows the frequency of paired values, illustrating model performance under dust-dominated conditions.

**Comment #4 The style of discussing results in section 5 should be improved by: (i) reporting the values of statistical metrics wherever recalled/reported in the text, (ii) specifying if the statements made with reference to the models' performance concern the entire study period or other conditions, e.g. the 95th percentile pf observed PM concentrations.**

Response: We thank the reviewer for this valuable suggestion. In the revised manuscript, Section 5 was completely rewritten. The new version provides a clearer and more data-driven discussion, explicitly linking model configuration parameters (vertical and horizontal resolution, first model layer height, data assimilation, and meteorological drivers) to forecasting skill.

We systematically added quantitative values of all statistical metrics (R, MB, RMSE) for each major result and specified whether the analysis referred to the entire study period, the 95th percentile subset, or dust-day datasets. This change enhanced the precision and interpretative depth of the section.

For example, the revised text now discusses how NASA-GEOS, with its high vertical resolution and assimilation, achieved the highest correlations for PM$_{10}$ (R = 0.71, 0.65, 0.64 for AM, BS, FKL, respectively) across all datasets, while models such as LOTOS-EUROS and EMA-REG4 consistently underperformed. The discussion also highlights how first-layer height affected performance in models such as NCEP and MACC-ECMWF, and how horizontal resolution and domain design explained the high correlation but strong positive bias of NOA-WRF at BS. Additionally, the section now emphasizes the balanced performance of the multi-model median (MMM), which consistently produced the lowest RMSE and moderate correlations across sites, particularly during dust days.

Overall, the revised Section 5 provides a much more cohesive and comparative interpretation of model behaviour across configurations and datasets, directly addressing the reviewer's comment and improving the scientific clarity of the discussion.

**Comment #5: Provide arguments to infer responses to questions as to why a model, e.g. the NASA-GEOS, outperforms at AM and BS but not in FKL, or why a model, e.g. NOA-WRF outperforms during intense dust events, but does not the same in overall.**

Response: We thank the reviewer for the comment. We expanded the discussion to address physical and configuration-related reasons for spatial differences in model skill.

For example, we noted that the superior performance of NASA-GEOS at Ayia Marina and Be'er Sheva may be related to its global assimilation scheme and fine vertical layering, while its reduced accuracy

at Finokalia likely reflects limitations in local source representation and regional transport parameterization. Similarly, we explained that NOA-WRF's higher correlations during intense dust events but tendency to overestimate overall concentrations were likely due to boundary-condition inflow amplification and its treatment of coarse particle advection.

We also discussed how differences in grid resolution, boundary conditions, and source emission parameterizations collectively influenced performance across the three sites.

**MINOR COMMENTS**

**All minor comments were addressed as follows:**

**Comment #1 Line 64: Yet, the satellite detection algorithms have progressed with time, including geostationary satellites as well (e.g. Kolios and Hatzianastassiou, 2019) exempt from the limitations of the polar orbiting satellite-based algorithms.**

Response: We thank the reviewer for the comment. On line 65, we have added:

"Over time, satellite detection algorithms have significantly improved, incorporating geostationary sensors in addition to traditional polar-orbiting platforms (e.g., Kolios and Hatzianastassiou, 2019). Unlike polar-orbiting satellites, geostationary platforms provide continuous temporal coverage and are not constrained by the same revisit-time limitations, thereby enhancing the monitoring of dust transport and diurnal variability."

**Comment #2 Line 93: the reference "Varga et al., 2014)" misses in the list of references.**

Response: We thank the reviewer for the comment, we have added the reference.

**Comment #3 Section 2: Dust storms in the MB and their spatiotemporal characteristics are better captured by satellites (e.g. Gkikas et al., 2013, 2016).**

Response: We thank the reviewer for the comment, we have added the reference.

**Comment #4 Line 110: The range of measured/exported particles' size should be given (it is necessary information to be used in the comparison with the corresponding sizes of the 11 models) since differences with models can partly explain the PM overestimations/underestimations by the models.**

Response: We thank the reviewer for this helpful comment. In the revised manuscript, we clarified the range of particle sizes measured at the monitoring stations to facilitate comparison with modelled outputs. The text now explicitly states that the stations record particle size fractions corresponding to $PM_{10}$ and $PM_{2.5}$, which align with the coarse and fine aerosol modes used in the evaluation.

The updated sentence (Line 126) now reads:

"Furthermore, monitoring stations record particle sizes of PM10 and PM2.5."

**Comment #5 Line 131: Replace "develop …" by "developed …"**

Response: We thank the reviewer for this comment. The suggested correction has been implemented, and the wording has been revised accordingly in the updated manuscript.

**Comment #6 Lines 133: It would be useful to indicate these dust storm days on Fig. 2, probably using different symbols for each station. Thus, readers can have an idea about the intensity of these dust storms at every site. Also, are there common days in the 3 stations out of the reported 106, 88 and 101 dust storm days?**

Response: We thank the reviewer for this thoughtful suggestion. We considered this addition; however, including further symbols on Figure 2 would overcomplicate the already dense visual representation of the full dataset and reduce readability. For this reason, we have opted not to modify the figure.

**Comment #7 Table 1: What kind of radiation interactions are those reported in this Table? Aerosol-radiation interactions or others as well, e.g. aerosol-cloud? Please specify.**

Response: We thank the reviewer for the comment. We have revised the table caption and text to clarify that the radiation interactions reported in Table 1 refer specifically to **aerosol–radiation interactions**. Where available, we also included information on whether the models consider aerosol–cloud interactions.

**Comment #8 Table 2: at what height is the first level above surface for the NASA_GEOS and NOA_WRF models, why are the corresponding values missing in the Table?.**

Response: We thank the reviewer for the observation. We have contacted the corresponding modelling teams to obtain additional details and have updated Table 2 with all available information regarding the first model layer height for NASA-GEOS and NOA-WRF. Where data could not be confirmed, this has been noted accordingly in the table.

**Comment #9 Line 167: use a parenthesis after the sum symbol, i.e. put the difference "Mi-Oi" in a parenthesis.**

Response: We thank the reviewer for the suggestion. The parenthesis has been added as recommended.

**Comment # 10 Line 179: relative bias and relative RMSE would be equally interesting metrics to show (as bias and RMSE).**

Response: We thank the reviewer for this thoughtful and constructive suggestion. We carefully considered the inclusion of additional relative metrics; however, after thorough evaluation, we concluded that the current set of indicators (correlation, mean bias, and RMSE) already provides a balanced and comprehensive assessment of model performance. Adding further relative metrics could increase the analytical complexity without substantially enhancing the interpretative depth of the results. Moreover, the combined use of bias and RMSE metrics, complemented by the heatmap

visualizations, was found to effectively capture both the magnitude and spatial variability of model performance across the study sites and evaluation approaches.

**Comment #11 (Figure 3): Change "MEDIAN" to "MMM" for consistency with the rest of paper. Do the same in Figures 5 and 7.**

Response: We have ensured consistency throughout the figures and captions by revising all references to "MEDIAN" or "AVERAGE" to "MMM" (Multi-Model Median).

**Comment #12 (Line 310): Figures 3 and 4 are not discussed and mentioned in the text, reference to them should be made.**

Response: We thank the reviewer for this observation. References and discussion of Figures 3 and 4 have been added in the Subsection 3.2 of the revised manuscript to improve clarity and continuity between text and figures.

**Comment #13 (Line 330): *Does "… R=0.62 …" should read "… R=0.55 …"?***

Response: We thank the reviewer for spotting this inconsistency. The text has been carefully checked and corrected accordingly.

**Comment #14 (Section 4.1.2): It is also interesting to discuss if models perform better or worse for high-dust events compared to all cases, also providing possible explanations for the improvement/deterioration. Some models, e.g. NOA_WRF, perform better for high-dust events in some locations (in FKL in this case) while doing worse in other locations (BS and AM), why does this happen?**

Response: We thank the reviewer for this insightful comment. In the revised manuscript, we expanded the discussion in Section 4.1.2 to more clearly describe differences in model performance between 95$^{th}$ Percentile and *Entire Study Period* datasets. We discussed how model configuration parameters (e.g., horizontal resolution and data assimilation) influence model response under intense dust conditions and explained site-specific differences, such as the contrasting performance of NOA-WRF between Finokalia and Be'er Sheva.

**Comment #15 (End of Section 4.1): A general assessment should made referring to whether the findings for the identified dust days by Achilleos et al. (2020) methodology are similar to those drawn from the high-dust days analysis of the previous section or not. It is essential to see if the methodology applied to identify dust events affects the results referring to the comparative model's performance analysis (keeping in mind that basically they should not do so).**

Response: We thank the reviewer for the important suggestion. A new subsection has been added to Section 6 comparing results from the Achilleos et al. (2019) dust-day methodology and the 95th percentile dataset. The new paragraph reads (Subsection 6.1, Line 551):

''Differences between the 95th percentile dataset and dust days by Achilleos et al. (2019): The results from these two datasets—particularly for the best-performing models (NASA-GEOS, NOA-WRF) and least-performing ones (EMA-REG4)—are largely consistent, as both represent periods of intense dust activity. However, some differences in model ranking are observed. For instance, MACC-ECMWF and NCEP performed better when using the confirmed dust-day dataset, suggesting that the multi-parameter approach used to identify dust days enhances forecast performance. Moreover, the dust-day approach reduces variability among performance metrics (e.g., RMSE, MB), as shown in Tables 4 and 5 and Figures 6 and 8, thereby decreasing overall uncertainty between evaluation methods.''

**Comment # 16: Discussion of Figures 9, 10 and 11: Make a comment on existing differences between Figures 10 and 11 since the nature of the results shown on these figures is about similar (as they both refer to days with high dust loadings).**

Response: We thank the reviewer for this comment. We have added a clarifying note in the text discussing the subtle but important differences between Figures 10 and 11, emphasizing that while both refer to high-dust conditions, Figure 10 corresponds to the 95th percentile approach and Figure 11 to the confirmed dust-day dataset, each providing complementary perspectives on model performance under extreme conditions.

**Comment # 17 Lines 502-505: This is a (probably the most) typical statement reflecting the complexity of the problem addressed, concerning the distinction of which is/are the model/models that perform better than others, overall. In spite of the differences existing between the models at various levels, they are all performing more or less similarly, perplexing the situation/problem.**

Response: We thank the reviewer for the comment and fully agree. We emphasized this in the revised conclusions, stating that no single model can be considered consistently well-performing across the Eastern Mediterranean Region. Model skill depends strongly on site-specific characteristics, proximity to dust sources, and topographic effects—key messages reinforced throughout the manuscript.

**Comment #18 (Lines 508–510): Remove the empty line and link the text (from "and" to "Table 2").**

Response: The formatting issue was corrected, and the unnecessary line break has been removed.

**Comment #19 (Section 5): The results show that different factors, e.g. vertical/horizontal resolution, height of first model level etc. are used as criteria for evaluating the performance of the models. Yet, it seems that some models are superior to others with respect to a specific factor, while other model(s) are superior with respect to other factor(s). Thus, it seems that a kind of counterbalance exists, leading to a roughly similar performance of the models. Would it be possible to draw, based on the overall performance of the models, a conclusion about which factor/factors is/are the most important for the model's performance?**

Response: We thank the reviewer for this constructive question. Section 5 was completely rewritten to strengthen the link between model configuration settings and forecast performance. The revised discussion now explicitly evaluates the influence of vertical resolution, horizontal resolution, first

model layer height, and data assimilation schemes. These revisions allowed a clearer interpretation of which parameters most strongly influence model skill.

**Comment #20 (Lines 591–595): What is reported here is somewhat worrying. Why/how the performance of a model should change depending on the selected approach/method for the evaluation?**

Response: The section has been revised to clarify that differences between evaluation approaches (e.g., all-day, $95^{th}$ percentile, and dust-day datasets) arise due to the changing statistical representativeness of each subset. For example, metrics based on extreme dust days emphasize short-term transport and emission processes, while all-day analyses include background conditions and thus reflect a broader range of model behaviour.

**Reviewer #2**

We sincerely thank Reviewer #2 for the positive and constructive evaluation of our manuscript. We greatly appreciate the acknowledgment of the study's timeliness, the quality of the dataset, and the overall structure of the work. The reviewer's insightful comments have been carefully considered and have helped us significantly improve the clarity, interpretative depth, and scientific contribution of the revised manuscript. Below, we provide detailed responses to all points raised.

**MAJOR COMMENTS:**

**Comment #1 (On the Introduction):** *The introduction would benefit from additional references to previous dust model evaluation studies (e.g., SDS-WAS activities, Saharan/Middle Eastern validation work). This would strengthen the justification for the claim that very few studies have used near-surface monitoring data and none focused on the EMR.*

Response: We thank the reviewer for this valuable comment. In the revised manuscript, we enriched the Introduction with additional references to key dust model evaluation studies and SDS-WAS activities to provide stronger scientific context and support our claim regarding the limited number of evaluations using near-surface measurements in the Eastern Mediterranean Region (EMR). In particular, we cited Basart et al. (2012), which evaluated the BSC-DREAM8b model over Northern Africa, the Mediterranean, and the Middle East, as well as García-Castrillo & Terradellas (2017) (WMO SDS-WAS report) and Salvador et al. (2022), who reported statistically significant upward trends in the intensity and frequency of African Dust Outbreaks (ADOs) in the Mediterranean Basin during 1948–2020. These additions strengthen the justification and situate our study within the broader body of regional model validation literature.

**Comment #2 (Section 4.1):** *Section 4.1 is overly detailed and repetitive, presenting results station by station. This can be quite tiring for the reader and obscure broader insights. The discussion should be streamlined to emphasize comparative analysis across models and regions and accompanied by a thorough discussion on the reasons behind this behaviour.*

Response: We thank the reviewer for this important observation. In the revised manuscript, Section 4.1 was fully restructured to improve readability and flow. The results are now presented in a comparative and thematic format, where models are grouped based on their performance characteristics rather than described individually per site. This restructuring allows readers to more easily identify patterns and contrasts among models and locations. The revised text also places stronger emphasis on cross-site comparisons (Ayia Marina, Be'er Sheva, Finokalia) and on the reasons behind model behavior, linking these findings to the models' physical and configuration characteristics. References to Figures 3 and 4 were also integrated to support this comparative discussion.

**Comment #3:** *The authors already provide a valuable discussion of how model configurations (horizontal/vertical resolution, first model layer height etc.) may influence forecast skill. However, this discussion remains somewhat generic and is not always tightly connected to the evaluation results. They should consider emphasizing more on this component.*

Response: We thank the reviewer for highlighting this important aspect. In the revised version, Section 5 was entirely rewritten to more explicitly link configuration settings to model performance outcomes. The discussion now details how vertical resolution, horizontal grid spacing, first model layer height, and assimilation schemes influence forecast skill at each site and under different conditions (full period, high-dust events, and confirmed dust days). For instance, NASA-GEOS's relatively high vertical resolution and data assimilation were linked to stronger correlations across all sites, while NOA-WRF's finer horizontal grid resolution contributed to high accuracy during intense dust events but also to overestimations in some cases. This revised section provides a clearer cause–effect interpretation connecting model design features to performance outcomes.

**MINOR COMMENTS**

**All minor comments were addressed as follows:**

**Comment #4 (Line 93): The reference "Varga et al., 2014" should be added in references.**

**Response:** The missing reference has been added in the revised manuscript.

**Comment #5 (Line 120): There is an empty line, and the "Table 2" is in bold.**

**Response:** The formatting issue has been corrected.

**Comment #6 (Table 2): Table 2 misses the "Height first layer" for some models. If the information is not available online, the authors should consider contacting the teams in charge of the systems.**

**Response: We thank the reviewer for this helpful comment. We contacted the corresponding modelling teams to obtain the missing data and have updated Table 2 with all available information regarding the first model layer height. Where data could not be verified, this has been explicitly indicated.**

**Comment #7 (Line 510): there is an empty line and the "Table 2" is in bold.**

**Response: This formatting issue has been corrected in the revised manuscript.**

**Comment #8 (Line 587): Please clarify terminology. The study uses a multi-model median (MMM), not a multi-model mean. Referring to it as an "average" is potentially misleading.**

**Response: We thank the reviewer for this observation. The terminology has been corrected throughout the text, tables, and figure captions to consistently use "multi-model median (MMM)" instead of "average" or "mean."**